# Ascent Fails to Forget

**Ioannis Mavrothalassitis**[*]   **Pol Puigdemont**[*]   **Noam Itzhak Levi**[*]   **Volkan Cevher**

LIONS, École Polytechnique Fédérale de Lausanne (EPFL), Lausanne, Switzerland
{ioannis.mavrothalassitis, pol.puigdemontplana, noam.levi}@epfl.ch

## Abstract

Contrary to common belief, we show that gradient ascent-based unconstrained optimization methods frequently fail to perform machine unlearning, a phenomenon we attribute to the inherent statistical dependence between the forget and retain data sets. This dependence, which can manifest itself even as simple correlations, undermines the misconception that these sets can be independently manipulated during unlearning. We provide empirical and theoretical evidence showing these methods often fail precisely due to this overlooked relationship. For random forget sets, this dependence means that degrading forget set metrics (which, for the oracle, should mirror test set metrics) inevitably harms overall test performance. Going beyond random sets, we consider logistic regression as an instructive example where a critical failure mode emerges: inter-set dependence causes gradient descent-ascent iterations to progressively diverge from the oracle. Strikingly, these methods can converge to solutions that are not only far from the oracle but are potentially even further from it than the original model itself, rendering the unlearning process actively detrimental. A toy example further illustrates how this dependence can trap models in inferior local minima, inescapable via finetuning. Our findings highlight that the presence of such statistical dependencies, even when manifest only as correlations, can be sufficient for ascent-based unlearning to fail. Our theoretical insights are corroborated by experiments on complex neural networks, demonstrating that these methods do not perform as expected in practice due to this unaddressed statistical interplay.

## 1 Introduction

Machine learning models have become an integral part of modern research and development methods, even in sensitive domains such as medicine, chemistry, and cybersecurity. This integration has led to growing concerns over data privacy and model maintenance. In this context, the process of selectively removing the influence of specific training examples from a trained model, namely machine *unlearning*, has emerged as a strongly desired capability [1]. Machine unlearning [2, 3] has garnered significant attention due to its diverse applications, ranging from addressing toxic or outdated data [4, 5], to resolving copyright concerns in generative models [6–8], and improving LLM alignment [9, 10].

The fundamental challenge in machine unlearning lies in designing efficient *unlearning algorithms* that do not degrade model performance.

Given a model $h_\theta$ with parameters $\theta$, trained on a dataset $\mathcal{D}$, and a subset $\mathcal{F} \subset \mathcal{D}$ to be forgotten, the goal of any unlearning algorithm is to produce a model $h_\theta^{\mathrm{UL}}$ that effectively simulates a model trained exclusively on the retain set $\mathcal{R} = \mathcal{D} \setminus \mathcal{F}$ [11]. While retraining from scratch on $\mathcal{R}$ provides a straightforward solution, it becomes computationally prohibitive on large datasets or as unlearning requests become more frequent.

---

[*]Equal contribution.

39th Conference on Neural Information Processing Systems (NeurIPS 2025).

For convex models, efficient unlearning algorithms with theoretical guarantees have been developed [12–17], which rely on variants of noisy descent algorithms (**no ascent steps**). However, due to the non-convex, non-smooth, and high-dimensional nature of deep neural network architectures, provable guarantees for unlearning are often lacking. Consequently, current methods frequently compromise model accuracy or require substantial modifications to training procedures [18, 19]. A notable recent exception is the rewind method for unlearning proposed by Mu and Klabjan [20], which provides guarantees for the unlearned model. However, this method is expensive, needing either substantial storage (to retain full model states from previous stages) or significant computational effort (due to the requirement of multiple proximal point iterations).

Many widely used and studied unlearning methods in practice [1, 21, 22] typically rely on fine-tuning heuristics to transform the initial model $h_\theta$ into an empirically unlearned model $\hat{h}_\theta^{\mathrm{UL}}$. The underlying idea of these methods is to reverse the effect that the forget set $\mathcal{F}$ has had on the model during training. Typically, these methods employ some variant of Gradient Ascent on forget set points and Gradient Descent on retain set points for a small number of fine-tuning epochs [23, 24]. We will refer to these methods as *Descent-Ascent (DA)* unlearning algorithms.

Unfortunately, recent evaluations and benchmarks demonstrate that DA approaches can be highly unreliable [25, 21, 26], as they neither possess theoretical performance guarantees nor clear mechanisms defining a stopping criterion for the unlearning process. Additionally, these methods are extremely sensitive to fine-tuning hyperparameters, most crucially the learning rate and the fine-tuning duration.

In this work, we identify an overlooked crucial obstacle for machine unlearning that is not taken into account by DA methods. Concretely, we show that the existence of data dependencies between samples in the forget and retain sets can lead to poor unlearning performance in some cases, as well as complete breakdown, even in convex settings.

Our main contributions can be summarized as follows:

1. We start by empirically showcasing that DA-based methods fail in practical settings under a robust evaluation and discuss limitations of previous methodologies.
2. Supported by our empirical findings, we first show theoretically that unlearning random forget sets is impossible without causing model degradation, as unlearning random sets is equivalent in distribution to unlearning samples from the population data distribution.
3. We move beyond forget and retain sets which share clear statistical dependencies to analyze the simple setting of multi-dimensional logistic regression, where we show inter-set correlations lead to DA failure modes.
4. In our logistic regression analysis, we differentiate the impact of DA unlearning based on forget set size. We specifically show that for certain forget set sizes, DA can be harmful to the model, even when employing arbitrary early stopping.
5. Finally, using low-dimensional examples, we demonstrate how DA can lead the model to suboptimal local minima, which do not align with the minima achieved through retraining.

**Notation:** We will use the following notation. We use uppercase bold letters for matrices $\boldsymbol{X} \in \mathbb{R}^{m \times n}$, lowercase bold letters for vectors $\boldsymbol{x} \in \mathbb{R}^m$ and lowercase letters for numbers $x \in \mathbb{R}$. Accordingly, the $i^{\mathrm{th}}$ row and the element in the $i, j$ position of a matrix $\boldsymbol{X}$ are given by $\boldsymbol{x}_i$ and $x_{ij}$ respectively. We use the shorthand $[n] = \{1, \cdots, n\}$ for any natural number $n$. Let $\mathbb{1}_{(\cdot,\cdot)} : \mathbb{R} \times \mathbb{R} \to \{0, 1\}$ such that $\mathbb{1}_{(x,x)} = 1$, otherwise for $x \neq y$, $\mathbb{1}_{(x,y)} = 0$. We will denote our model with parameters $\theta$ as $h_\theta : \mathbb{R}^d \to \mathbb{R}$. We define a training dataset of size $|\mathcal{D}|$ as a set of samples and labels $\{(\boldsymbol{x}_i, y_i)\}_{i=1}^{|\mathcal{D}|} = \mathcal{D}$, composed of a "retain" set $\mathcal{R}$ and "forget" set $\mathcal{F}$ such that $|\mathcal{D}| = |\mathcal{R}| + |\mathcal{F}|$. We take "ascent" optimization on a sample to mean computing the gradient update w.r.t. to a loss $\nabla_\theta \ell$ and flipping its sign when updating the model parameters.

## 2 Related Work

**Machine unlearning:** Unlearning methods can be used to either remove particular samples [11, 1, 27, 12, 18], or to remove subsets of the data which share certain underlying features, captured by abstract concepts [28–31]. In this work, we focus on the prior, though we believe some of our results may be extended to the latter setting. Exact unlearning methods [18] offer theoretical guarantees but often sacrifice accuracy, leading to widespread adoption of approximate methods in deep learning. These approximate approaches are evaluated through membership inference attacks [22, 32, 25] and

backdoor removal capabilities [4]. As Thudi et al. [33] note, meaningful evaluation must focus on algorithmic behavior rather than individual models due to deep learning's stochastic nature. For a review of open problems in machine unlearning, see [34] and references therein.

**Unlearning approaches in deep learning:** Current approaches primarily use gradient-based methods, including partial fine-tuning [32], AD combinations [21], and sparsity-regularized fine-tuning [35]. Alternative methods employ local quadratic approximations [22, 36] or influence functions [37]. One of the most used unlearning methods, SCRUB [25] fine-tunes models using KL divergence objectives, but faces similar underlying challenges as other methods. The approach presented in Georgiev et al. [38] introduces a predictive data attribution approach with good unlearning quality under a robust evaluation, although it raises some scalability concerns if we account for the full cost the method. In this work we focus on DA based methods. Georgiev et al. [38] also observe empirical failures of Gradient Descent/Ascent style updates in certain settings. We attribute these failures to inter-set correlation between forget and retain data, and contribute controlled experiments disentangling random from structurally correlated forget sets, together with theory in non-linear models showing first-step detriment of DA independent of early stopping; see App. B.

## 3 Ascent Methods Fail in the Wild

To evaluate the quality of unlearning rigorously, we adopt the KLoM (KL Divergence of Margins [38]) metric, which quantifies the distributional difference in predictions between the unlearned model and the oracle model. KLoM measures the KL-Divergence between the classifier margin distributions of 100 unlearned models against 100 Oracle models. A KLoM score approaching zero, the lowest possible, indicates near-perfect unlearning.

In our main experiments, we examine two gradient-based unlearning approaches which are commonly used as baselines: Gradient Ascent (GA) which performs steps in the direction of the gradients of the model on forget set, and Gradient Descent/Ascent (GDA) which adds descent steps on the retain set after the initial ascent steps, for each epoch. We conduct these experiments using ResNet-9 models on Cifar-10 [39] under forget sets of different sizes and properties. Fig. 1 illustrates our results on a selected forget set. We observe that both GA and GDA methods either fail to substantially move away from the pretrained initialization or severely degrade model performance. Our choices in model, dataset, forget sets and results are consistent with the values reported in Georgiev et al. [38].

These outcomes highlight an important limitation in the empirical evaluation of GA based unlearning methods. It is necessary for a hyperparameter selection criteria to be defined, ideally, before deploying the method or at least without measuring at the final target metric. It is not fair to do an instance-specific selection of the best run after having seen the evaluation due to bias. For a small enough forget set and a large enough grid of runs with different hyperparameters we could trick ourselves into a false sense of unlearning even with vanilla GA. This problem is showcased in Fig. 2 and is rooted in the missing targets problem [25, 38], which amounts to the difficulty of not having a target stopping value for GA based unlearning optimization procedures. On top of that, different points seem to unlearn at different rates [38] which suggests that such a stopping value would need to be point-specific.

We also observe that the difficulty of unlearning varies greatly depending on the specific forget set selected, as shown in Fig. 3. In general, we find GA and GDA methods to be fragile. The extreme sensitivity to hyperparameters, unclear stopping criteria for Gradient Ascent, and substantial computational costs in using Gradient Descent on the retain set to fix models, severely restrict their practicality. Fundamentally, performing gradient ascent on individual points is not aligned with the core definition of unlearning, making these approaches unsuitable for reliable and consistent machine unlearning in real-world scenarios. In the Appendix, we include the methodology details for forget sets, KLoM, hyperparameters along with additional results on more forget sets, models (ResNet-18 [40]) and datasets (ImageNetLiving-17 [41, 42]).

Motivated by these results, the following sections aim to demonstrate that the underlying statistical data dependencies may be a central cause for the typical failure modes of DA based unlearning methods, both in general, and in some useful tractable settings.

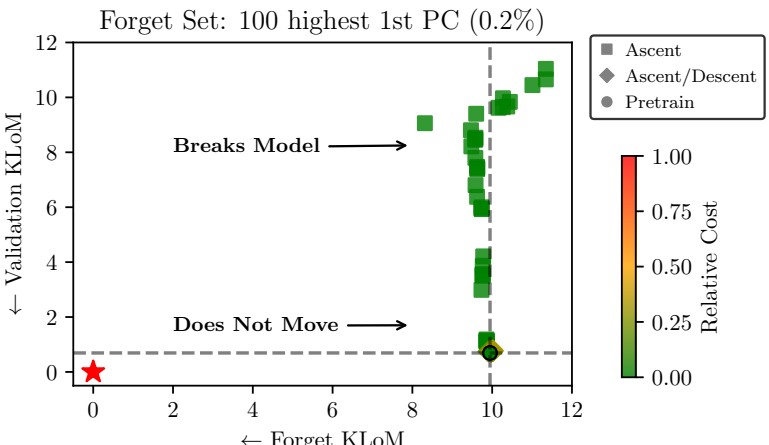

Figure 1: **Ascent Fails to Forget.** We apply Gradient Ascent and Gradient Descent/Ascent to Pretrained models to unlearn a selected forget set containing points of the first Principal Component (PC) of the influence matrix from Cifar-10. KLoM scores (x-axis, y-axis) measure the quality of unlearning on a given set by comparing the distribution distance between unlearned predictions and Oracle predictions (0 means perfect unlearning ★). We measure KLoM values over each data-point in a set and report the 95th percentile in each group. Different (x/y) points in the plot represent results for different unlearning method hyper-parameters. The colors indicate what is the relative cost of an unlearning method when compared to fully retraining the model. A Pretrained model (○) is similar to an Oracle on the validation set but very different on the forget set. On such set, unlearning with Gradient Ascent or Gradient Descent/Ascent either breaks the model or does not move much from the Pretrained starting point, we find this behavior to be consistent in most sets. Forget set selection and KLoM score metric follow Georgiev et al. [38]. Further details on method and evaluation hyper-parameters can be found in the Appendix.

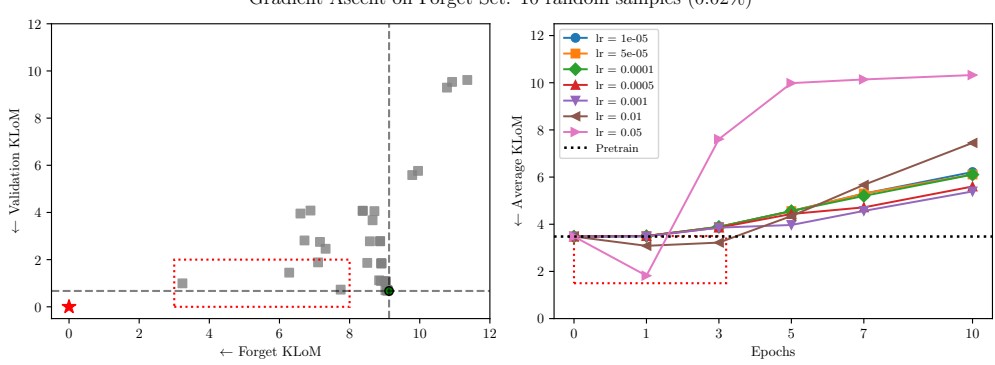

Figure 2: **The Ascent Forgets Illusion.** The left plot shows KLoM scores of Gradient Ascent when unlearning just 10 random samples (axis and points follow Fig. 1). Some runs (- - -) seem to achieve unlearning without breaking the model. On the right, we present the average KLoM between retain, validation and forget sets (y-axis) along time of unlearning (x-axis). We observe that in order for Gradient Ascent to unlearn such (easy) sets in practice, one would need to (i): select the learning rate, (ii) know when to stop fine-tuning.

## 4 Unlearning and Random Sets

A natural starting point for understanding how data correlations influence the unlearning process is that of random forget sets. If a forget set is selected uniformly at random from the original set, it is evident that the two sets would have high statistical dependence between them. Therefore, we would naturally expect that metrics measured in the forget set would be indistinguishable to those of the test set and very close to those of the retain for the oracle. We can state this formally in Lemma 1 for

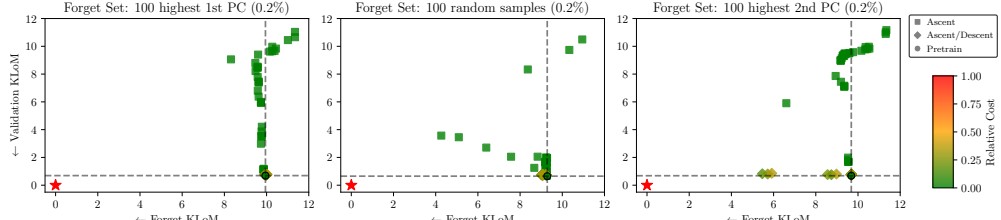

Figure 3: **Different Unlearning Difficulties.** We present the KLoM scores of Gradient Ascent and Gradient Descent/Ascent when unlearning over different forget sets (axes and points follow Fig. 1). In general, the majority of runs either do nothing or break the model. Empirically, we find highly important points (left) to be the hardest to unlearn with zero realizations showing any unlearning signs at all. Random samples (center) show some Gradient Ascent runs improving the forget KLoM but with significant degradation in the models. Finally, for a set with second PC points (right) we observe some Gradient Descent/Ascent runs improve the forget KLoM without breaking the model but at a high cost, around $25\%$ of retraining an Oracle for unlearning $0.2\%$ of the data.

accuracy; the same reasoning extends to other metrics (e.g., regression losses) after modifying the metric definition. The proof of Lemma 1 can be found in App. C.

**Lemma 1** (Random Sets). *Given a true distribution of samples $P_{\mathcal{T}}$ and a forget set $\mathcal{F}$ chosen uniformly at random from the dataset and a oracle model with parameters $\theta$, then the probability that the accuracy on the test set $Acc_{\mathcal{T}}$ and the forget set $Acc_{\mathcal{F}}$ diverge from one another by more than $\epsilon$ is upperbounded by the following inequality:*

$$P\left(|Acc_{\mathcal{T}} - Acc_{\mathcal{F}}| \geq \epsilon\right) \leq 2\exp\left(-2|\mathcal{F}|\epsilon^2\right).$$

Lemma 1 suggests that any unlearning method which successfully approximates the oracle should result in a model which performs equally well on both the forget set and test set. Therefore, an unlearned model with poor forget set performance will statistically diverge from an oracle model, whose forget set accuracy reflects the accuracy of the test set and is high for modern machine learning tasks. Consequently, methods based on DA that explicitly degrade a metric on the forget set might be more harmful rather than beneficial (this critique does not apply to non-Gradient Descent/Ascent approaches such as influence-function or certified-unlearning methods, which do not aim to worsen the forget-set metric). This raises the following question:

*(I) Do data dependencies, in general, cause DA methods to have a detrimental effect on the models, without actually unlearning?*

While the answer to this question under statistical dependencies is apparently positive, as stated in Lemma 1, when considering limited data dependencies, the answer becomes more convoluted. Before proceeding to the discussion regarding this, we would like to point out that a "good" unlearning algorithm should not be harmful to the model regardless of the input forget set $\mathcal{F}$, even for random sets.

In the following sections, we analyze several tractable scenarios. We find that in many cases the answer to (I) is positive, implying that data dependencies do, in fact, cause DA methods to have a detrimental effect on the models.

## 5 Models Diverge from Retraining Solutions Under DA Unlearning

We begin our study with logistic regression in a high-dimensional, nearly orthogonal setting where correlations are only between samples on the same dimension. We then generalize to cross-dimensional correlations, and finally study a nonlinear example in low-dimensions on a small fixed dataset.

### 5.1 High Dimensions: Correlated Data Causes Diverging Solutions in Logistic Regression

Here, we study the problem of binary logistic regression with a ridge parameter $\lambda$, and weights $\mathbf{w}$ on nearly orthogonal data in $d$ dimensions. Based on the work of Soudry et al. [43], we use the exponential loss $\ell_i = e^{y_i h_\theta(\mathbf{x}_i)}$ as a more tractable proxy for the logistic loss. The pre-training ($\mathcal{D}$),

retraining ($\mathcal{R}$) and GDA optimization methods (DA) will minimize their respective losses

$$\mathcal{L}_{\mathcal{D}} = \frac{1}{|\mathcal{D}|} \sum_{i=1}^{\mathcal{D}} e^{-y_i \cdot \langle \mathbf{w}, \mathbf{x}_i \rangle} + \frac{\lambda}{2} \|\mathbf{w}\|_2^2, \qquad \mathcal{L}_{\mathcal{R}}, = \frac{1}{|\mathcal{R}|} \sum_{i=1}^{\mathcal{R}} e^{-y_i \cdot \langle \mathbf{w}, \mathbf{x}_i \rangle} + \frac{\lambda}{2} \|\mathbf{w}\|_2^2,$$

$$\mathcal{L}_{\mathrm{DA}} = \frac{1}{|\mathcal{R}|} \sum_{i=1}^{\mathcal{R}} e^{-y_i \cdot \langle \mathbf{w}, \mathbf{x}_i \rangle} - \frac{1}{|\mathcal{F}|} \sum_{i=1}^{\mathcal{F}} e^{-y_i \cdot \langle \mathbf{w}, \mathbf{x}_i \rangle} + \frac{\lambda}{2} \|\mathbf{w}\|_2^2. \tag{1}$$

### 5.1.1 Data Correlations on a Single Dimension

We start from the case of a semi orthogonal dataset. Using the following assumptions:

**Assumption 1.** *The data is separable into orthogonal sets $S_j$ for each coordinate $j$. More specifically, after jointly permuting samples and coordinates, the data matrix is block diagonal: each block corresponds to samples $S_j$ that have nonzero entries only on a disjoint coordinate set $C_j$. Samples from different sets are orthogonal; within a set, samples may be correlated.*

**Assumption 2.** *For a coordinate $j$ it holds that for all samples $i$ with $x_{i,j} \neq 0$, $y_i \cdot x_{i,j} = 1$.*

These assumptions correspond to a dataset in $d$ dimensions where there are sets of samples on orthogonal axes to one another. As a result, data points that lie in different sets $S_j$ are perfectly orthogonal and uncorrelated; however, data points that lie in the same set may be correlated with one another.

Recall our hypothesis that data dependencies can cause DA methods to degrade model metrics, instead of converging to an oracle model, we will pick a subset of a set $S_j$ as our forget set. This will allow for a simplistic analysis while testing the hypothesis for a highly correlated forget set.

Let $|\mathcal{R}_j|$ the size of the retain set for samples with $x_{i,j} \neq 0$, then in order to model the behavior of the minimizers of Eq. (1), for forget sets of different sizes, we define the $j$th forget set fraction size as $|\mathcal{F}_j| = \alpha \cdot |\mathcal{R}_j|$. A simple example of this setting can be a set of retain points of $x_j = 1, y_j = 1$ and a set of forget points of $x_j = -1, y_j = -1$, where we are practically requested to remove all (or some) of the negative samples. The effect of unlearning a forget set on a particular coordinate axis $j$, can then be shown to obtain closed form solutions as given by Lemma 2, proven in App. D.2.

**Lemma 2** (Closed Form). *Let $w_j^{\mathcal{D}}, w_j^{\mathcal{R}}$ and $w_j^{DA}$ be the $j^{th}$ coordinate of **any** local minima/maxima for the logistic regression problems defined in Eq. (1), then they admit the form:*

$$w_j^{\mathcal{D}} = \mathrm{W}\left(\frac{(1+\alpha)|\mathcal{R}_j|}{\lambda|\mathcal{D}|}\right), w_j^{\mathcal{R}} = \mathrm{W}\left(\frac{|\mathcal{R}_j|}{\lambda|\mathcal{R}|}\right), w_j^{DA} = \mathrm{W}\left(\frac{(1-\alpha|\mathcal{R}|/|\mathcal{F}|)|R_j|}{\lambda|R|}\right),$$

*where $W(z)$ corresponds to the Lambert-W function, the solution to $z = W(z)e^{W(z)}$.*

It follows directly from Lemma 2, that by changing the value of $\alpha$, which determines the ratio of the size of the forget set to that of the retain in this coordinate, the solutions will be ordered by their magnitude. Concretely, Lemma 3 shows that the DA solution is always **farthest** away from the oracle solution. For example, in 1D, "farthest away" means that $(w_j^{DA} - w_j^{\mathcal{D}})$ and $(w_j^{\mathcal{R}} - w_j^{\mathcal{D}})$ have opposite signs, so $w_j^{DA}$ and $w_j^{\mathcal{R}}$ lie on opposite sides of the pre-trained solution $w_j^{\mathcal{D}}$; therefore any step toward $w_j^{DA}$ moves the model away from the oracle $w_j^{\mathcal{R}}$. Meanwhile, the oracle and pre-trained solutions remain close and more importantly the DA solution and the oracle solution lie in opposite directions with respect to the initial solution of pre-training $w_j^{\mathcal{D}}$. This observation implies that performing DA in this setup always converges away from the oracle solution, thus doing nothing at all is a better strategy than DA. The aforementioned observation can be formally decomposed in the following lemmas. We prove Lemma 3 in App. F, which gives a formal statement regarding the fact that the minima of DA and the oracle are in opposite directions with respect to the minimum of the intial dataset $\mathcal{D}$.

**Lemma 3** (Divergence Logistic Regression). *Let $w_j^{\mathcal{D}}, w_j^{\mathcal{R}}$ and $w_j^{DA}$ the $j^{th}$ coordinate of the convergence point for the logistic regression problem for the original set $\mathcal{D}$, the retain set $\mathcal{R}$ and the Descent Ascent method respectively. Then for a range of $\alpha$ we have that: $\left(w_j^{DA} - w_j^{\mathcal{D}}\right) \cdot \left(w_j^{\mathcal{D}} - w_j^{\mathcal{R}}\right) \geq 0$.*

We defer the reader to App. F for the exact range of $\alpha$, for which Lemma 3 holds, let us point out that the lemma holds for $\alpha \leq |\mathcal{F}|/|\mathcal{R}|$, this means that if we were working on a purely 1 dimensional

dataset, this lemma would **always** hold. Lemma 3 answers our original question of whether data correlations cause DA methods to harm the model in the positive. Before proceeding to the study of higher dimensions, we would like to comment on the stability of the process of unlearning under DA methods.

**Stability of DA methods:** We begin by characterizing the distance between the different stationary points for the three problems.

Lemma 4 provides an upperbound on the distance between the oracle solution and the initial solution for $\mathcal{D}$. Its counterpart, Lemma 5 provides a lower bound on the distance between the oracle solution and the DA solution. The proof for Lemma 4 can be found in App. G, while the proof for Lemma 5 lies in App. H

**Lemma 4** (Distance Growth). *Let $w_j^{\mathcal{D}}, w_j^{\mathcal{R}}$ the $j^{th}$ coordinate of the convergence point for the logistic regression problem for the original set $\mathcal{D}$ and the retain set $\mathcal{R}$ respectively. It holds that the distance $\Delta_{\mathcal{R},\mathcal{D}} = |w_j^{\mathcal{D}} - w_j^{\mathcal{R}}| \leq \left|\ln\left((1+\alpha)\frac{|\mathcal{R}|}{|\mathcal{D}|}\right)\right|$, for any $\lambda > 0$ and $\alpha > 0$.*

**Lemma 5** (Distance Unlearning). *Let $w_j^{\mathcal{R}}, w_j^{DA}$ the $j^{th}$ coordinate of the convergence point for the logistic regression problem for the retain set $\mathcal{R}$ and the Descent Ascent method respectively. It holds that for for $\alpha \geq |\mathcal{F}|/|\mathcal{R}|$ the distance $\Delta_{\mathcal{R},DA} = |w_j^{\mathcal{R}} - w_j^{DA}| \geq W_0\left(|\mathcal{R}_j|/(\lambda|\mathcal{R}|)\right)$*

Employing Lemma 4 and Lemma 5, one can derive the following Corollary.

**Corollary 1.** *As the ridge $\lambda \to 0$ for $\alpha \to |\mathcal{F}|/|\mathcal{R}|$, we have that $\Delta_{\mathcal{R},\mathcal{D}} \to 0$ and $\Delta_{\mathcal{R},DA} \to \infty$.*

Cor. 1 demonstrates how unlearning using DA is very volatile and even a few steps of the method can cause the model to diverge.

**A possible stabilization effect of iterative DA:** So far, we have focused on the behavior of minimizers of Eq. (1), which describes a simultaneous descent-ascent algorithm. In practice, however, iterative methods are typically used, where one first performs a step of ascent on the forget set, followed descent on the retain set. In App. D.3, we show that for small learning rates $\eta \to 0$, the iterative method is nearly identical to the simultaneous update. Namely the derivative used for the update rule is

$$w_j^{t+1} \leftarrow w_j^t - \eta\left(-\frac{|\mathcal{R}_j|}{|\mathcal{R}|}e^{-w_j^t} + \frac{\alpha \cdot |\mathcal{R}_j|}{|\mathcal{F}|}e^{-w_j^t} + 2\lambda w_j^t\right),$$

where the only difference is a factor of 2 in front of the regularization that differs from the normal DA loss. We have omitted a term which is of the order of $\mathcal{O}\left(\eta^2\right)$, since the solution, should it exist has $w_j^t$ small and a term with $\eta^2 \to 0$ has negligable contribution.

The leading correction term $\mathcal{O}\left(\eta^2\right)$ which was omitted in the update rule above stops the algorithms solution $w_j^{\text{DA}}$ from diverging, since the term is of the form

$$\eta^2\alpha\frac{|\mathcal{R}_j|^2}{|\mathcal{F}||\mathcal{R}|}e^{-2w_j^t} - \eta^2\lambda w_j^t\alpha\frac{|\mathcal{R}_j|}{|\mathcal{F}|}e^{-w_j^t},$$

which increases for larger $w_j^t$. This addresses stability concerns; however, it does nothing to remedy our main concern raised in Lemma 3 regarding the harmful effect of these methods on the model.

### 5.1.2 Cross Dimensional Data Correlations

In the previous section we studied the case where our samples are fully correlated, since they existed in a single dimension. In this section we will consider the two dimensional case where we have two sets of samples $S_i$ and $S_j$, which have values $x_i = (0,\ldots,0,1,\epsilon,0,\ldots,0)$ and $x_j = (0,\ldots,0,\epsilon,1,0,\ldots,0)$ respectively. We will consider the case where the samples of $S_i$ are all in the retain set, while the samples of $S_j$ are all in the forget. In this case the correlation between the samples in the forget and the retain set depends on $\epsilon$ and therefore this allows us to do a parametric study of the effect of correlation between the forget and the retain on the performance of DA based methods. In similar fashion to the 1 dimensional case we will consider that the forget set $|\mathcal{F}_{i,j}| = \alpha|\mathcal{R}_{i,j}|$, where $F_{i,j}, R_{i,j}$ the forget and the retain over the $i,j$ dimensions, respectively. In order to facilitate the analysis we will change the coordinate system only for the $i$ and the $j$ coordinate

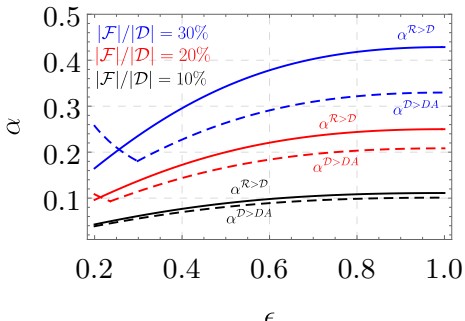

Figure 4: **Cross dimensional data correlations $\epsilon$ lead DA to failure for a certain range of values.** We present the range of $\alpha$ as a function of the correlation $\epsilon$, for which we can guarantee that DA is detrimental. The (- -) lines represent the minimum $\alpha$ for which the coordinates of the original model become bigger than the coordinates of the DA unlearning algorithm and with the (–) the maximum $\alpha$ for which the coordinates of the oracle are bigger than those of the original model.

to $x = w_i + \epsilon w_j$ and $y = w_i \epsilon + w_j$. Let $x^\mathcal{R}, y^\mathcal{R}$ the coordinates for the oracle model stationary point, $x^\mathcal{D}, y^\mathcal{D}$ for the pretrain model and $x^{\mathrm{DA}}, y^{\mathrm{DA}}$ for the DA unlearning scheme, we can give the following characterizations:

**Lemma 6.** *The closed form solution for the stationary points for the retrain set is given as:*

$$x^\mathcal{R} = \mathrm{W}\left(\frac{(1+\epsilon^2)|R_{i,j}|}{\lambda|\mathcal{R}|}\right), \quad y^\mathcal{R} = \frac{2\epsilon}{1+\epsilon^2}\mathrm{W}\left(\frac{(1+\epsilon^2)|R_{i,j}|}{\lambda|\mathcal{R}|}\right).$$

**Lemma 7.** *For the stationary points of the original set, one can derive the following ranges.*

$$\mathrm{W}\left(\frac{|R_{i,j}|}{\lambda|\mathcal{D}|}((1+\epsilon^2)+2\alpha\epsilon)\right) \le x^\mathcal{D} \le \frac{2\epsilon}{1+\epsilon^2}W\left(\frac{\alpha(1+\epsilon^2)|R_{i,j}|}{\lambda|\mathcal{D}|}\right) + \mathrm{W}\left(\frac{(1+\epsilon^2)|R_{i,j}|}{\lambda|\mathcal{D}|}\right),$$

$$\mathrm{W}\left(\frac{\alpha(1+\epsilon^2)|R_{i,j}|}{\lambda|\mathcal{D}|}\right) \le y^\mathcal{D} \le \mathrm{W}\left(\frac{|R_{i,j}|}{\lambda|\mathcal{D}|}(2\epsilon+\alpha(1+\epsilon^2))\right).$$

**Lemma 8.** *For the stationary points of the model trained by DA methods we can derive the following ranges.*

$$x^{DA} \le \mathrm{W}\left(\frac{|R_{i,j}|}{\lambda|\mathcal{R}|}(1+\epsilon^2) - \frac{|R_{i,j}|}{\lambda|\mathcal{F}|}\alpha 2\epsilon\right), \qquad y^{DA} \le \mathrm{W}\left(\frac{|R_{i,j}|}{\lambda|\mathcal{R}|}2\epsilon - \frac{|R_{i,j}|}{\lambda|\mathcal{F}|}\alpha(1+\epsilon^2)\right).$$

While the problem becomes more complex in this case and to our knowledge it is not possible to compute an exact solution, the above Lemmas provide enough information for our purpose. The proofs for all of these Lemmas can be found in App. I.1. In similar fashion to the 1 dimensional case we would like to show that there exists a reasonable $\alpha$, for which we have that $(x^\mathcal{R}-x^\mathcal{D})\cdot(x^\mathcal{D}-x^{\mathrm{DA}}) \ge 0$ and at the same time $(y^\mathcal{R}-y^\mathcal{D})\cdot(y^\mathcal{D}-y^{\mathrm{DA}}) \ge 0$.

**Lemma 9.** *For $\alpha \ge \alpha^{\mathcal{D}>DA} = \max\left\{\frac{1+\epsilon^2}{2\epsilon}\frac{|\mathcal{F}|^2}{|\mathcal{R}|(|\mathcal{D}|+|\mathcal{F}|)}, \frac{2\epsilon}{1+\epsilon^2}\frac{|\mathcal{F}||\mathcal{D}|}{|\mathcal{R}|(|\mathcal{D}|+|\mathcal{F}|)}\right\}$ we have that $x^\mathcal{D} \ge x^{DA}$ and that $y^\mathcal{D} \ge y^{DA}$.*

**Lemma 10.** *For $\alpha \le \alpha^{\mathcal{R}>\mathcal{D}} = \min\left\{\alpha_x^{\mathcal{R}>\mathcal{D}}, \alpha_y^{\mathcal{R}>\mathcal{D}}\right\}$ we have that $x^\mathcal{R} \ge x^\mathcal{D}$ and that $y^\mathcal{R} \ge y^\mathcal{D}$, with $\alpha_x^{\mathcal{R}>\mathcal{D}}, \alpha_y^{\mathcal{R}>\mathcal{D}}$.*

We omit the exact values of $\alpha_x^{\mathcal{R}>\mathcal{D}}$ and $\alpha_y^{\mathcal{R}>\mathcal{D}}$, which can be found in App. I.2 along with the proofs for Lemma 9 and Lemma 10. Since the range of $\epsilon$ for which $(x^\mathcal{R}, y^\mathcal{R}) \ge (x^\mathcal{D}, y^\mathcal{D}) \ge (x^{\mathrm{DA}}, y^{\mathrm{DA}})$ cannot be resolved analytically, we show numerically in Fig. 4 that this range is typically large, and broadens as the fraction of samples to be forgotten increases, while the relevant window of correlation strength $\epsilon$ is wider for smaller correlation.

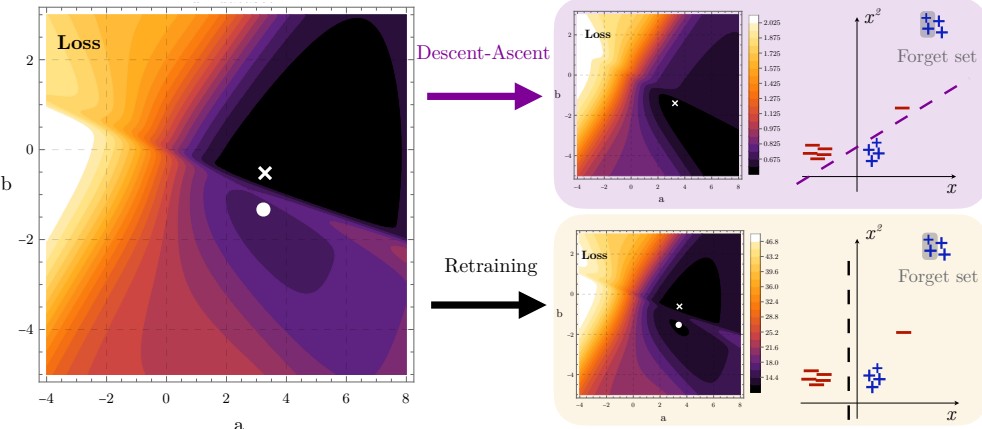

Figure 5: **Unlearning certain forget sets leads to the wrong decision boundary under GDA.**
*Left:* We show the MSE loss landscape for a pretrained model on the problem described in Sec. 5.2.
We denote as (×) the global minimum, while (○) is the local minimum. *Right:* The effective loss
landscape observed in the GDA problem (top) and the retraining problem (bottom). The combination
of these results shows that retraining keeps the model in the same global optimum as the pretrained
model, while GDA chooses the local minimum. This is clearly manifest in the decision boundaries
favored by the different methods, denoted in dashed lines. Next to the contour plots we present two
dimensional illustrations of possible decision boundaries between the samples labeled as negative
(−) and positive (+), while the forget set are the two positive points shaded in gray, as described in
Sec. 5.2. We show the decision boundaries for both GDA (right top) and retraining (right bottom).
These conclusions concern the minimizers of a fixed objective and thus do not depend on training
dynamics (e.g., step size); see App. B for details.

## 5.2   Low Dimensions: Descent-Ascent Favors The Wrong Solutions

While our previous theoretical analysis demonstrates that DA methods can be harmful to the model,
it fails to demonstrate a final concern about these methods, we would like to raise. *Is it possible to
remedy the harmful effects of these methods through finetuning on the retain afterwards?*

The answer that we give to this question unfortunately is not always, for neural networks or in general
non-convex function classes. To demonstrate this let us consider a binary classification problem
using a two dimensional kernel, with labels $y_i \in \{-1, 1\}$, data composed of $\mathbf{x}_i = (x_i, x_i^2)$ and
Mean Squared Error (MSE) loss with ridge regularization $\lambda \in \mathbb{R}^+$. The network is taken to be a
sigmoidal network with two parameters $\theta = (a, b)$, such that its output is $h_\theta(\mathbf{x}_i) = \sigma(ax_i + bx_i^2)$,
where $\sigma(z) = 1/(1 + e^{-(1+z)/2})$.

We choose 4 samples in the configuration: $\mathcal{D} = \{\mathbf{x}_1, \mathbf{x}_2, \mathbf{x}_3, \mathbf{x}_4\} = \{(-1, 1), (1, 1), (3, 9), (4, 16)\}$,
with labels $\{y_1, y_2, y_3, y_4\} = \{-1, 1, -1, 1\}$, respectively. In order to model the effect of multiple
points clustered together, we give each point a different weight in the loss function, such that

$$\mathcal{L} = \frac{1}{|\mathcal{D}|} \sum_{i=1}^{|\mathcal{D}|} \alpha_i \ell_i + \frac{\lambda}{2} \|\theta\|_2^2, \tag{2}$$

where $\ell_i$ are the single sample loss functions, and $\{\alpha_1, \alpha_2, \alpha_3, \alpha_4\} = \{5, 4, 1, 4\}$ represent the
number of points clustered together, as illustrated in Fig. 5, where $\lambda = 0.1$. This means that the
effective number of points that the classifier sees is $\sum_i \alpha_i$. The data configuration is chosen to
illustrate the failure mode of DA, while the dataset selection is arbitrary the key mode of failure is the
high correlation between the forget set and a subset of the retain.

Suppose we would like to unlearn two of the positive samples positioned at $\mathbf{x}_4$. Retraining would
correspond to simply setting $\alpha_4 = 2$, and applying gradient descent. Notice that this provides little
to no change for the minima location and the contour lines between the original dataset $\mathcal{D}$ and the
retraining set $\mathcal{R}$. In contrast, Performing GDA would amount to setting $\alpha_4 = 0$, since two points will
contribute the exact opposite gradient as the other two at the same position, effectively erasing them.

We find that this example can be simply understood by counting arguments: since the original dataset contains effectively 6 negative samples and 8 positive samples, the optimal decision boundary is given by the separating plane which correctly classifies the largest number of samples.

The pretrained model is optimal when $x_1$, $x_2$ and $x_4$ are correctly classified, while mislabeling $x_3$ (13 correct, 1 incorrect). Retraining simply reduces the weight of $x_4$, and keeping the same plane is still preferential (11 correct, 1 incorrect). However, performing GDA sets the gradients of half of the points at $x_4$ to cancel the other half, so it optimal to re-orient the decision boundary so that all samples are correctly classified (10 correct, 0 incorrect), while in reality, the algorithm has been tricked into finding a suboptimal solution (10 correct, 2 incorrect).

The qualitative analysis of this two-dimensional example shows that certain choices of forget sets that are highly correlated to the retain can lead to irreversible model degradation when using DA.

## 6 Conclusions

While our findings highlight significant challenges in current ascent-based unlearning methods, we believe that they are instructive for the construction of safer future methods. The weaknesses we identify primarily stem from ascent disregarding the data dependencies between the forget and the retain set. Future research on ascent based methods should take these dependencies into consideration. Our findings also suggests that methods based on rewinding [20] or stochastic methods based on noise [44] can be valid alternative schemes when being agnostic on the dataset properties.

## Acknowledgements

We would like to thank Roy Rinberg and Gal Vardi for insightful discussions.

This work was supported by Hasler Foundation Program: Hasler Responsible AI (project number 21043). Research was sponsored by the Army Research Office and was accomplished under Grant Number W911NF-24-1-0048. This work was funded by the Swiss National Science Foundation (SNSF) under grant number 200021_205011. Any opinions, findings, and conclusions or recommendations expressed in this material are those of the authors and do not necessarily reflect the views of the sponsors.

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

# A Limitations

**Limitations:** A key limitation in our theoretical results is their simplicity, both in the model analyzed as well as the methods, for which the analysis is done. The proof does not explicitly prohibit more complex models or methods from resolving this issue. We believe, however, that as far as models go simpler models could be nested in more complex ones, leading to this detrimental phenomenon. For developing more complex methods based on ascent we believe that still someone has to take correlations into consideration, given our findings. Extending our theoretical results to multi-layer networks would be valuable but is technically challenging. Quantifying which correlations are most harmful remains open. We propose using an influence function cross influence matrix between forget and retain sets as a tractable proxy, and testing whether Gradient Descent/Ascent unlearning degrades as its spectral norm increases.

# B Discussion

**Q: Do training dynamics (e.g., learning rate and regularization) affect the locations of the optimal minimizers and the resulting decision boundaries in Fig. 5 and Sec. 5.2?**
**A:** No. Our statements concern the optimization landscape of a fixed objective and the set of its minimizers. Training dynamics (including learning rate and optimizer trajectory) affect the path and speed of convergence, not which points are minimizers, as long as the objective is unchanged. Adjusting the ridge parameter shifts minimizers radially (toward/away from the origin) without altering their relative positions; hence the decision boundaries in Fig. 5 and the conclusions in Sec. 5.2 are unaffected. That said, using different learning rates for the retain and forget sets (i.e., the ascent and descent steps on DA) effectively changes the objective itself. In this case, the new objective becomes implicitly weighted by the ratio of those learning rates. While one could tune the weighting so that the particular case in Figure 5 no longer appears problematic, a similar counterexample can always be constructed for any ratio. For instance, if the forget set is updated with twice the step size of the retain set, then in the setting of Figure 5 one can simply reduce the forget set from two samples to one (by removing one of the original samples). This modification recreates the same optimization landscape we described earlier.

**Q: What results do you obtain when working with highly correlated datasets such as MNIST or FashionMNIST?**
**A:** MNIST [45] is degenerate for evaluating unlearning: many points are near-duplicates, so pretrained and oracle models remain nearly indistinguishable on forget points, yielding uniformly low KLoM across splits even when the forget set is 10% of training. In contrast to CIFAR-10, where oracles and pretrained models diverge on forget sets (consistent with Georgiev et al. [38]), MNIST shows little separability. While FashionMNIST [46] exhibits a modest increase in forget-set KLoM it is still far below CIFAR-10 magnitudes. As a result, any method may appear to succeed on MNIST by doing very little (for example, tiny steps), making true forgetting difficult to verify.

| Dataset | Forget % | Forget KLoM (95th) | Retain KLoM (95th) | Val KLoM (95th) |
|---------|----------|--------------------|--------------------|-----------------|
| MNIST | 0.02% | 0.5 | 0.7 | 0.71 |
| MNIST | 0.2% | 2.72 | 2.88 | 2.9 |
| MNIST | 10% | 1.79 | 1.65 | 1.71 |
| FashionMNIST | 0.02% | 4.13 | 1.87 | 2.76 |
| FashionMNIST | 0.2% | 2.72 | 1.79 | 2.70 |
| FashionMNIST | 10% | 2.69 | 2.0 | 3.31 |

Table 1: 95th percentile KLoM comparing pretrained and oracle models on MNIST and FashionM-NIST across forget-set sizes. Averages over 100 pretrained models per dataset and 100 oracles per forget set. **MNIST and FashionMNIST are highly correlated and therefore degenerate for sound machine unlearning evaluation** with pretrained and oracle models remaining too similar on forget points, which can inflate apparent success.

**Q: Why does Gradient Descent/Ascent unlearning degrade more when the forget set is aligned with the top principal components (1st/2nd PCs) compared to random forget sets?**
**A:** Intuitively, aligning the forget set with the top principal components (PCs) concentrates it along

directions where the model is most sensitive, so ascent steps taken to degrade performance on that set inevitably interfere with the decision boundary more globally. A simple 2-D logistic regression toy helps illustrate this. Let labels be generated by a fixed teacher vector $T \in \mathbb{R}^2$, e.g., $T = (1, -1)$, which classifies points by the sign of $x_1 - x_2$. Train a student with weights $S \in \mathbb{R}^2$ on these labels; successful learning yields $S \approx T$. If points are sampled near the origin, the (pointwise) influence of a training point $z = (x, y)$ on a test point $z_{\text{test}} = (x_{\text{test}}, y_{\text{test}})$ can be approximated (up to an inverse Hessian factor that is proportional to the identity) by

$$I_{z,z_{\text{test}}} \approx -y_{\text{test}} y\, \sigma\!\left(-y_{\text{test}}\, S^\top x_{\text{test}}\right) \sigma\!\left(-y\, S^\top x\right) x_{\text{test}}^\top x,$$

which, for $S \approx (1, -1)$, reduces to

$$I_{z,z_{\text{test}}} \approx -(x_{\text{test},1} - x_{\text{test},2})(x_1 - x_2)\, \sigma\!\left(-(x_{\text{test},1} - x_{\text{test},2})^2\right) \sigma\!\left(-(x_1 - x_2)^2\right) x_{\text{test}}^\top x.$$

Along lines of roughly constant $x_1 - x_2$, the correlation between $x$ and $x_{\text{test}}$ controls influence magnitude, so the leading PCs of $x_{\text{test}}^\top x$ align with the leading directions of the influence matrix. Selecting the forget set along the top PCs thus targets directions that most strongly affect predictions, causing ascent to push the model in ways that globally perturb the decision boundary. In contrast, random forget sets spread mass across many weaker directions, so ascent tends to be less coherent and, on average, less damaging. We stress this is an intuition; analyzing full influence matrices in tractable yet realistic settings would be valuable future work.

**On prior empirical observations and our contributions.** Georgiev et al. [38] empirically show that gradient ascent–based unlearning can perform poorly. For instance, low stability and different points unlearning at different rates and ascent diverging for linear models (Figs. 5 and 11 in [38]). We advance these observations by identifying inter-set correlation between forget and retain data as the causal mechanism and by providing the missing evidence: controlled experiments that disentangle random forget sets (with unstructured dependence) from structurally correlated sets aligned with top principal components of the influence matrix, and a theory for non-linear models showing immediate detriment of DA regardless of early stopping. Empirically, we find DA to be unstable on random sets sometimes improving the forget metric but often breaking the model. On structurally correlated sets, failure is systematic and severe (Figs. 1 and 3). We also highlight the evaluation pitfall where selecting the best run over large hyperparameter grids can create a false appearance of success (the "Ascent Forgets Illusion," Fig. 2). Theoretically, in logistic regression we show that DA moves away from the oracle from the very first steps: Lemma 3 proves that DA updates point in the opposite direction to the oracle, and Lemma 5 shows the DA solution can remain far from the oracle even when the retrained solution is close to the pretrained one. Together, these results explain *why* ascent–descent updates degrade performance in correlated regimes and *when* such degradation is unavoidable.

**Practical guidance and evaluation checklist.** We recommend two simple diagnostics for unlearning experimentation: (i) measure the oracle–pretrained KLoM gap on forget versus retain/validation splits and (ii) probe sensitivity to early stopping and step size, which can create the "Ascent Forgets Illusion".

# C  Proof of Lemma for Random Sets

In this section we provide proof that for a forget set, selected uniformly at random from the dataset it is with high probability impossible to differentiate the accuracy, loss, or any other metric between the test and the forget set, given that both of them are large enough. In this section we provide the proof for the accuracy metric, but for other metrics the proof follows in like manner. Intuitively this stems from the fact that for a model which has "unlearned" a forget set, that set is a random set for it.

We will use the following notation. Let $\mathbb{1}_{(\cdot,\cdot)} : \mathbb{R} \times \mathbb{R} \to \{0, 1\}$ such that $\mathbb{1}_{(x,x)} = 1$, otherwise for $x \neq y$, $\mathbb{1}_{(x,y)} = 0$. We will denote our model with parameters $\theta$ as $h_\theta : \mathbb{R}^d \to \mathbb{R}$.

**Lemma 1** (Random Sets). *Given a true distribution of samples $P_\mathcal{T}$ and a forget set $\mathcal{F}$ chosen uniformly at random from the dataset and a oracle model with parameters $\theta$, then the probability that the accuracy on the test set $Acc_\mathcal{T}$ and the forget set $Acc_\mathcal{F}$ diverge from one another by more than $\epsilon$ is upperbounded by the following inequality:*

$$P\left(|Acc_\mathcal{T} - Acc_\mathcal{F}| \geq \epsilon\right) \leq 2 \exp\left(-2|\mathcal{F}|\epsilon^2\right).$$

*Proof.* For each sample $(x_i, y_i)$, we calculate the correct response on that sample, as $\mathbb{1}_{(h_\theta(x_i), y_i)}$, consequently the response of the model for any sample is an independent rendom variable. So we get the following random variables, which correspond to the accuracy of the model on the forget set $\mathcal{F}$ and the test set $\mathcal{T}$ respectively.

$$\text{Acc}_\mathcal{T} = \mathbb{E}_{(x_i, y_i) \sim P_\mathcal{T}} \left[ \mathbb{1}_{(h_\theta(x_i), y_i)} \right]$$

$$\text{Acc}_\mathcal{F} = \frac{1}{|\mathcal{F}|} \sum_{(x_i, y_i) \in \mathcal{F}} \mathbb{1}_{(h_\theta(x_i), y_i)}$$

In order to proceed we will utilize Hoeffding's Inequality, which we state below for completeness:

**Lemma 11.** *Let $Z_1, Z_2, \ldots, Z_n$ be independent random variables such that $Z_i \in [a_i, b_i]$. Define their sum as:*

$$S_n = \sum_{i=1}^{n} Z_i$$

*and let $\mathbb{E}[S_n]$ be the expected value of $S_n$. Then, for any $t > 0$, the following bound holds:*

$$P\left(|S_n - \mathbb{E}[S_n]| \geq nt\right) \leq 2 \exp\left(\frac{-2n^2 t^2}{\sum_{i=1}^{n}(b_i - a_i)^2}\right)$$

In our case we have that $\frac{1}{n} S_n = \text{Acc}_\mathcal{F}$. Since the Forget set $\mathcal{F}$ is selected uniformly at random, we have that:

$$
\begin{aligned}
\mathbb{E}[\text{Acc}_\mathcal{F}] &= \mathbb{E}\left[ \frac{1}{|\mathcal{F}|} \sum_{(x_i, y_i) \in \mathcal{F}} \mathbb{1}_{(h_\theta(x_i), y_i)} \right] \\
&= \frac{1}{|\mathcal{F}|} \sum_{(x_i, y_i) \in \mathcal{F}} \mathbb{E}_{(x_i, y_i) \sim P_\mathcal{T}} \left[ \mathbb{1}_{(h_\theta(x_i), y_i)} \right] \\
&= \mathbb{E}_{(x_i, y_i) \sim P_\mathcal{T}} \left[ \mathbb{1}_{(h_\theta(x_i), y_i)} \right] \\
&= \text{Acc}_\mathcal{T}
\end{aligned}
$$

Since the random variables $\mathbb{1}_{(h_\theta(x_i), y_i)} \in [0, 1]$, we have that:

$$P\left(|\text{Acc}_\mathcal{T} - \text{Acc}_\mathcal{F}| \geq \epsilon\right) \leq 2exp\left(-2|\mathcal{F}|\epsilon^2\right)$$

which gives the lemma statement. $\qquad\square$

The above lemma gives a formal statement, as to why maximizing the error on random forget sets does not correspond to true unlearning, since the metrics in the forget set should match those in the test set.

## D  Logistic Regression

### D.1  Problem Statement

The logistic regression problem for the full dataset $\mathcal{D}$, retain set $\mathcal{R}$ and for the Descent-Ascent algorithm can be restated as:

$$\textbf{minimization } \mathcal{D}: \frac{1}{|\mathcal{D}|} \sum_{i=1}^{\mathcal{D}} e^{-y_i \cdot \langle w, x_i \rangle} + \frac{\lambda}{2} \|w\|_2^2$$

$$\textbf{minimization } \mathcal{R}: \frac{1}{|\mathcal{R}|} \sum_{i=1}^{\mathcal{R}} e^{-y_i \cdot \langle w, x_i \rangle} + \frac{\lambda}{2} \|w\|_2^2 \tag{3}$$

$$\textbf{Descent } \mathcal{R} - \textbf{Ascent } \mathcal{F}: \frac{1}{|\mathcal{R}|} \sum_{i=1}^{\mathcal{R}} e^{-y_i \cdot \langle w, x_i \rangle} - \frac{1}{|\mathcal{F}|} \sum_{i=1}^{\mathcal{F}} e^{-y_i \cdot \langle w, x_i \rangle} + \frac{\lambda}{2} \|w\|_2^2$$

## D.2 Single Dimension

In this section, we compare the solutions of training a logistic regression model on a full dataset $\mathcal{D}$, purely on the retain set $\mathcal{R}$ and doing GDA on the forget set $\mathcal{F}$. We will also include a regularization term. The corresponding objective functions would be:

We can derivate the above to get the following equations for their solutions respectively.

$$\text{(minimization } \mathcal{D}) \qquad \frac{1}{|\mathcal{D}|}\sum_{i=1}^{\mathcal{D}} -y_i \cdot x_i e^{-y_i \cdot \langle w, x_i \rangle} + \lambda w = 0$$

$$\text{(minimization } \mathcal{R}) \qquad \frac{1}{|\mathcal{R}|}\sum_{i=1}^{\mathcal{R}} -y_i \cdot x_i e^{-y_i \cdot \langle w, x_i \rangle} + \lambda w = 0$$

$$\text{(Descent } \mathcal{R} - \text{Ascent } \mathcal{F}) \quad \frac{1}{|\mathcal{R}|}\sum_{i=1}^{\mathcal{R}} -y_i \cdot x_i e^{-y_i \cdot \langle w, x_i \rangle} - \frac{1}{|\mathcal{F}|}\sum_{i=1}^{\mathcal{F}} -y_i \cdot x_i e^{-y_i \cdot \langle w, x_i \rangle} + \lambda w = 0$$

So we can express each coordinate $j$ of the minimizer for the three cases, as:

$$\text{(minimization } \mathcal{D}) \qquad w_j = \frac{1}{\lambda|\mathcal{D}|}\left(\sum_{i=1}^{\mathcal{D}} y_i \cdot x_{i,j} e^{-y_i \cdot \langle w, x_i \rangle}\right)$$

$$\text{(minimization } \mathcal{R}) \qquad w_j = \frac{1}{\lambda|\mathcal{R}|}\left(\sum_{i=1}^{\mathcal{R}} y_i \cdot x_{i,j} e^{-y_i \cdot \langle w, x_i \rangle}\right)$$

$$\text{(Descent } \mathcal{R} - \text{Ascent } \mathcal{F}) \quad w_j = \frac{1}{\lambda|\mathcal{R}|}\left(\sum_{i=1}^{\mathcal{R}} y_i \cdot x_{i,j} e^{-y_i \cdot \langle w, x_i \rangle}\right) - \frac{1}{\lambda|\mathcal{F}|}\left(\sum_{i=1}^{\mathcal{F}} y_i \cdot x_{i,j} e^{-y_i \cdot \langle w, x_i \rangle}\right)$$

## D.3 Iterating Gradient Descent and Ascent

Here, we consider the iterative gradient descent-ascent algorithm, where we first perform a gradient descent step on the retain set, followed by a gradient ascent step on the forget set. We show that to leading order in the small learning rate expansion, the solution found by iterative GA is identical to the one given by GA in Eq. (3). For iterative GA, the dynamics are given by

$$w_j^{t+1} = w_j^t + \eta\left(\frac{|\mathcal{R}_j|}{|\mathcal{R}|}e^{-w_j^t} - \lambda w_j^t\right), \tag{4}$$

$$w_j^{t+2} = w_j^{t+1} - \eta\left(\frac{\epsilon \cdot |\mathcal{R}_j|}{|\mathcal{F}|}e^{-w_j^{t+1}} + \lambda w_j^{t+1}\right),$$

where $\eta$ is the learning rate for both steps. Plugging in the result of $w_j^{t+1}$ into the expression for $w_j^{t+2}$ and expanding for small $\eta \ll 1$, we obtain the following update rule

$$w_j^{t+2} = w_j^t + \eta\left(\frac{|\mathcal{R}_j|}{|\mathcal{R}|}e^{-w_j^t} - \lambda w_j^t\right) \tag{5}$$

$$- \eta\left(\frac{\epsilon \cdot |\mathcal{R}_j|}{|\mathcal{F}|}e^{-\left(w_j^t + \eta\left(\frac{|\mathcal{R}_j|}{|\mathcal{R}|}e^{-w_j^t} - \lambda w_j^t\right)\right)} + \lambda\left(w_j^t + \eta\left(\frac{|\mathcal{R}_j|}{|\mathcal{R}|}e^{-w_j^t} - \lambda w_j^t\right)\right)\right)$$

$$\simeq w_j^t + \eta\left(\frac{|\mathcal{R}_j|}{|\mathcal{R}|}e^{-w_j^t} - 2\lambda w_j^t\right) - \eta\left(\frac{\epsilon \cdot |\mathcal{R}_j|}{|\mathcal{F}|}e^{-\left(w_j^t + \eta\left(\frac{|\mathcal{R}_j|}{|\mathcal{R}|}e^{-w_j^t} - \lambda w_j^t\right)\right)}\right)$$

$$\simeq w_j^t + \eta\left(\frac{|\mathcal{R}_j|}{|\mathcal{R}|}e^{-w_j^t} - 2\lambda w_j^t\right) - \eta\left(\frac{\epsilon \cdot |\mathcal{R}_j|}{|\mathcal{F}|}e^{-w_j^t}\left(1 - \eta\left(\frac{|\mathcal{R}_j|}{|\mathcal{R}|}e^{-w_j^t} - \lambda w_j^t\right)\right)\right)$$

$$= w_j^t - \eta\left(-\frac{|\mathcal{R}_j|}{|\mathcal{R}|}e^{-w_j^t} + \frac{\epsilon \cdot |\mathcal{R}_j|}{|\mathcal{F}|}e^{-w_j^t} + 2\lambda w_j^t\right) + \mathcal{O}(\eta^2).$$

Eq. (5) shows that up to order $\mathcal{O}(\eta^2)$, the dynamics, as well as the convergent solution of the iterative descent-ascent algorithm are identical to the ones obtained from Eq. (3), up to a rescaling of the regularization parameter by a factor of 2, as in $\lambda_{\text{DA}} = 2\lambda_{\text{Iter}-\text{DA}}$.

# E  Proof of Lemma 2

In this section we prove Lemma 2 under Assumption 1 and Assumption 2.

**Lemma 2** (Closed Form). *Let $w_j^{\mathcal{D}}, w_j^{\mathcal{R}}$ and $w_j^{DA}$ be the $j^{th}$ coordinate of **any** local minima/maxima for the logistic regression problems defined in Eq. (1), then they admit the form:*

$$w_j^{\mathcal{D}} = \mathrm{W}\left(\frac{(1+\alpha)|\mathcal{R}_j|}{\lambda|\mathcal{D}|}\right), w_j^{\mathcal{R}} = \mathrm{W}\left(\frac{|\mathcal{R}_j|}{\lambda|\mathcal{R}|}\right), w_j^{DA} = \mathrm{W}\left(\frac{(1-\alpha|\mathcal{R}|/|\mathcal{F}|)|R_j|}{\lambda|R|}\right),$$

*where $W(z)$ corresponds to the Lambert-W function, the solution to $z = W(z)e^{W(z)}$.*

*Proof.* Let us start by restating the original problem as given in Eq. (3). For the sake of completeness.

**minimization $\mathcal{D}$ :** $\dfrac{1}{|\mathcal{D}|}\displaystyle\sum_{i=1}^{\mathcal{D}} e^{-y_i\cdot\langle w,x_i\rangle} + \dfrac{\lambda}{2}\|w\|_2^2$

**minimization $\mathcal{R}$ :** $\dfrac{1}{|\mathcal{R}|}\displaystyle\sum_{i=1}^{\mathcal{R}} e^{-y_i\cdot\langle w,x_i\rangle} + \dfrac{\lambda}{2}\|w\|_2^2$

**Descent $\mathcal{R}$ − Ascent $\mathcal{F}$ :** $\dfrac{1}{|\mathcal{R}|}\displaystyle\sum_{i=1}^{\mathcal{R}} e^{-y_i\cdot\langle w,x_i\rangle} - \dfrac{1}{|\mathcal{F}|}\displaystyle\sum_{i=1}^{\mathcal{F}} e^{-y_i\cdot\langle w,x_i\rangle} + \dfrac{\lambda}{2}\|w\|_2^2$

We can get the local minima of these functions by using Fermat's theorem, therefore we have:

**minimization $\mathcal{D}$ :** $\dfrac{1}{|\mathcal{D}|}\displaystyle\sum_{i=1}^{\mathcal{D}} -y_i\cdot x_i e^{-y_i\cdot\langle w,x_i\rangle} + \lambda w = 0$

**minimization $\mathcal{R}$ :** $\dfrac{1}{|\mathcal{R}|}\displaystyle\sum_{i=1}^{\mathcal{R}} -y_i\cdot x_i e^{-y_i\cdot\langle w,x_i\rangle} + \lambda w = 0$

**Descent $\mathcal{R}$ − Ascent $\mathcal{F}$ :** $\dfrac{1}{|\mathcal{R}|}\displaystyle\sum_{i=1}^{\mathcal{R}} -y_i\cdot x_i e^{-y_i\cdot\langle w,x_i\rangle} - \dfrac{1}{|\mathcal{F}|}\displaystyle\sum_{i=1}^{\mathcal{F}} -y_i\cdot x_i e^{-y_i\cdot\langle w,x_i\rangle} + \lambda w = 0$

Solving the equations for coordinate $j$ and using Assumption 1, we get:

**minimization $\mathcal{D}$ :** $w_j = \dfrac{1}{\lambda|\mathcal{D}|}\left(\displaystyle\sum_{i=1}^{S_j} y_i\cdot x_{i,j} e^{-y_i\cdot w_j\cdot x_{i,j}}\right)$

**minimization $\mathcal{R}$ :** $w_j = \dfrac{1}{\lambda|\mathcal{R}|}\left(\displaystyle\sum_{i=1}^{\mathcal{R}_j} y_i\cdot x_{i,j} e^{-y_i\cdot w_j\cdot x_{i,j}}\right)$

**Descent $\mathcal{R}$ − Ascent $\mathcal{F}$ :** $w_j = \dfrac{1}{\lambda|\mathcal{R}|}\left(\displaystyle\sum_{i=1}^{\mathcal{R}_j} y_i\cdot x_{i,j} e^{-y_i\cdot w_j\cdot x_{i,j}}\right) - \dfrac{1}{\lambda|\mathcal{F}|}\left(\displaystyle\sum_{i=1}^{\mathcal{F}_j} y_i\cdot x_{i,j} e^{-y_i\cdot w_j\cdot x_{i,j}}\right)$

Now we can utilize Assumption 2 and the fact that: $|\mathcal{F}_j| = \alpha\cdot|\mathcal{R}_j|$ to restate the previous equations in the form:

**minimization $\mathcal{D}$ :** $w_j = \dfrac{(1+\alpha)|\mathcal{R}_j|}{\lambda|\mathcal{D}|}e^{-w_j}$

**minimization $\mathcal{R}$ :** $w_j = \dfrac{|\mathcal{R}_j|}{\lambda|\mathcal{R}|}e^{-w_j}$

**Descent $\mathcal{R}$ − Ascent $\mathcal{F}$ :** $w_j = \dfrac{|\mathcal{R}_j|}{\lambda|\mathcal{R}|}e^{-w_j} - \dfrac{\alpha\cdot|\mathcal{R}_j|}{\lambda|\mathcal{F}|}e^{-w_j}$

As explained in App. E.1 the Lambert function W provides the solution for equations of the previous form. Using this fact we get:

**minimization $\mathcal{D}$ :** $w_j^{\mathcal{D}} = \mathrm{W}\left(\dfrac{(1+\alpha)|\mathcal{R}_j|}{\lambda|\mathcal{D}|}\right)$

**minimization $\mathcal{R}$ :** $w_j^{\mathcal{R}} = \mathrm{W}\left(\dfrac{|\mathcal{R}_j|}{\lambda|\mathcal{R}|}\right)$

**Descent $\mathcal{R}$ − Ascent $\mathcal{F}$ :** $w_j^{\mathrm{DA}} = \mathrm{W}\left(\dfrac{(1-\alpha|\mathcal{R}|/|\mathcal{F}|)|R_j|}{\lambda|R|}\right)$

This concludes the proof. $\qquad\square$

### E.1 The Lambert function $W$

In this section for the sake of exposition we briefly discuss the Lambert function $W$. Introduced by Johann Heinrich Lambert in 1758. In this work we are primarily interested in the property of the function that for any $\alpha$, the solution of the equation:

$$x - \alpha \cdot e^{-x} = 0$$

is $x = W(-a)$. As well as the monotonicity of the principal branch of the Lambert function.

## F  Proof of Lemma 3

In this section of the appendix we provide the proof for Lemma 3, under Assumptions 1 and 2, we start by restating the Lemma below for the sake of exposition.

**Lemma 3** (Divergence Logistic Regression). *Let $w_j^{\mathcal{D}}, w_j^{\mathcal{R}}$ and $w_j^{DA}$ the $j^{th}$ coordinate of the convergence point for the logistic regression problem for the original set $\mathcal{D}$, the retain set $\mathcal{R}$ and the Descent Ascent method respectively. Then for a range of $\alpha$ we have that: $\left(w_j^{DA} - w_j^{\mathcal{D}}\right) \cdot \left(w_j^{\mathcal{D}} - w_j^{\mathcal{R}}\right) \geq 0$.*

*Proof.* To begin the proof let us restate the three minimization problems for logistic regression for the three cases, whose respective solutions are $w_j^{\mathcal{D}}, w_j^{\mathcal{R}}, w_j^{DA}$

$$\textbf{minimization } \mathcal{D}: \frac{1}{|\mathcal{D}|} \sum_{i=1}^{\mathcal{D}} e^{-y_i \cdot \langle w, x_i \rangle} + \frac{\lambda}{2} \|w\|_2^2$$

$$\textbf{minimization } \mathcal{R}: \frac{1}{|\mathcal{R}|} \sum_{i=1}^{\mathcal{R}} e^{-y_i \cdot \langle w, x_i \rangle} + \frac{\lambda}{2} \|w\|_2^2$$

$$\textbf{Descent } \mathcal{R} - \textbf{Ascent } \mathcal{F}: \frac{1}{|\mathcal{R}|} \sum_{i=1}^{\mathcal{R}} e^{-y_i \cdot \langle w, x_i \rangle} - \frac{1}{|\mathcal{F}|} \sum_{i=1}^{\mathcal{F}} e^{-y_i \cdot \langle w, x_i \rangle} + \frac{\lambda}{2} \|w\|_2^2$$

So the local minima and maxima of these equations can be characterized with the help of Lemma 2, the proof of which can be found in App. E, for the sake of completeness, let us restate the lemma here

**Lemma 2** (Closed Form). *Let $w_j^{\mathcal{D}}, w_j^{\mathcal{R}}$ and $w_j^{DA}$ be the $j^{th}$ coordinate of **any** local minima/maxima for the logistic regression problems defined in Eq. (1), then they admit the form:*

$$w_j^{\mathcal{D}} = W\left(\frac{(1+\alpha)|\mathcal{R}_j|}{\lambda|\mathcal{D}|}\right), w_j^{\mathcal{R}} = W\left(\frac{|\mathcal{R}_j|}{\lambda|\mathcal{R}|}\right), w_j^{DA} = W\left(\frac{(1-\alpha|\mathcal{R}|/|\mathcal{F}|)|R_j|}{\lambda|R|}\right),$$

*where $W(z)$ corresponds to the Lambert-W function, the solution to $z = W(z)e^{W(z)}$.*

Since $\alpha \geq 0$, we have that:

$$\frac{(1+\alpha)|\mathcal{R}_j|}{\lambda|\mathcal{D}|} > 0 \text{ and } \frac{|\mathcal{R}_j|}{\lambda|\mathcal{R}|} > 0$$

The minimization for logistic regression over the original dataset $\mathcal{D}$ and the retrain dataset $\mathcal{R}$ both have a global minimum that is unique and corresponds to the solution of the principal branch of the Lambert function $W_0$, for that value.

For the Descent Ascent solution, since the input of the Lambert function is not necessarily positive, we have to separate our analysis to three cases:

1. The first case, where there is only one global minimum, meaning that the input $x$ of the Lambert function is $x \geq 0$. Equivalently, we have $\frac{(1-\alpha|\mathcal{R}|/|\mathcal{F}|)|R_j|}{\lambda|R|} \geq 0$ which implies that $\alpha \leq \frac{|\mathcal{F}|}{|\mathcal{R}|}$

2. The second case, where we have a solution both for the primary and the secondary branch of the Lambert function, corresponding to a local maximum and minimum respectively meaning that you have that the input $x$ of the Lambert function is $-1/e \leq x \leq 0$, equivalently solving for $\epsilon$ gives $|\mathcal{F}|/|\mathcal{R}| < \alpha \leq |\mathcal{F}|/|\mathcal{R}| + (\lambda|\mathcal{F}|)/(e|\mathcal{R}_j|)$

3. The third case, where there are no local minima, meaning that the input of the Lambert function $x$ is $x < -1/e$, which implies that $\alpha > |\mathcal{F}|/|\mathcal{R}| + (\lambda|\mathcal{F}|)/(e|\mathcal{R}_j|)$

**Case 1:** In case 1 we have that $\alpha \leq \frac{|\mathcal{F}|}{|\mathcal{R}|}$, which implies that:

$$\frac{(1+\alpha)|\mathcal{R}_j|}{\lambda|\mathcal{D}|} \leq \frac{(|\mathcal{R}| + |\mathcal{F}|)|\mathcal{R}_j|}{\lambda|\mathcal{R}||\mathcal{D}|} \leq \frac{|\mathcal{D}||\mathcal{R}_j|}{\lambda|\mathcal{R}||\mathcal{D}|} \leq \frac{|\mathcal{R}_j|}{\lambda|\mathcal{R}|}$$

so since the principal branch $W_0$ of the Lambert function is increasing, we have that:

$$w_j^{\mathcal{D}} = W_0\left(\frac{(1+\alpha)|\mathcal{R}_j|}{\lambda|\mathcal{D}|}\right) \leq W_0\left(\frac{|\mathcal{R}_j|}{\lambda|\mathcal{R}|}\right) = w_j^{\mathcal{R}}$$

For this case, let us now assume that $\alpha \geq |\mathcal{F}|^2/\left(|\mathcal{R}|(|\mathcal{F}| + |\mathcal{D}|)\right)$, it is easy to verify that for such an $\alpha$ it holds that: $\frac{(1+\alpha)|\mathcal{R}_j|}{\lambda|\mathcal{D}|} \geq \frac{(1-\alpha|\mathcal{R}|/|\mathcal{F}|)|\mathcal{R}_j|}{\lambda|R|}$, so we have that:

$$w_j^{\text{DA}} = W_0\left(\frac{(1 - \alpha|\mathcal{R}|/|\mathcal{F}|)|\mathcal{R}_j|}{\lambda|R|}\right) \leq W_0\left(\frac{(1+\alpha)|\mathcal{R}_j|}{\lambda|\mathcal{D}|}\right) = w_j^{\mathcal{D}}$$

So for **Case 1** we have that $w_j^{\text{DA}} \leq w_j^{\mathcal{D}} \leq w_j^{\mathcal{R}}$, which implies that $(w_j^{\text{DA}} - w_j^{\mathcal{D}}) \cdot (w_j^{\mathcal{D}} - w_j^{\mathcal{R}}) \geq 0$

This concludes the proof.

$\square$

## G Proof of Lemma 4

In this section we provide the proof for Lemma 4 under Assumptions 1 and 2.

**Lemma 4** (Distance Growth). *Let $w_j^{\mathcal{D}}, w_j^{\mathcal{R}}$ the $j^{th}$ coordinate of the convergence point for the logistic regression problem for the original set $\mathcal{D}$ and the retain set $\mathcal{R}$ respectively. It holds that the distance $\Delta_{\mathcal{R},\mathcal{D}} = |w_j^{\mathcal{D}} - w_j^{\mathcal{R}}| \leq \left|\ln\left((1+\alpha)\frac{|\mathcal{R}|}{|\mathcal{D}|}\right)\right|$, for any $\lambda > 0$ and $\alpha > 0$.*

*Proof.* We start from Lemma 2, which we restate below for the sake of exposition.

**Lemma 2** (Closed Form). *Let $w_j^{\mathcal{D}}, w_j^{\mathcal{R}}$ and $w_j^{DA}$ be the $j^{th}$ coordinate of **any** local minima/maxima for the logistic regression problems defined in Eq. (1), then they admit the form:*

$$w_j^{\mathcal{D}} = W\left(\frac{(1+\alpha)|\mathcal{R}_j|}{\lambda|\mathcal{D}|}\right), w_j^{\mathcal{R}} = W\left(\frac{|\mathcal{R}_j|}{\lambda|\mathcal{R}|}\right), w_j^{DA} = W\left(\frac{(1 - \alpha|\mathcal{R}|/|\mathcal{F}|)|R_j|}{\lambda|R|}\right),$$

*where $W(z)$ corresponds to the Lambert-W function, the solution to $z = W(z)e^{W(z)}$.*

Since the input of the Lambert function for $w_j^{\mathcal{D}}, w_j^{\mathcal{R}}$ is always positive these solutions correspond to the only minimum of the function for the minimization problem and additionally they are calculated from them principal branch of the Lambert function $W_0$. We start from the logarithmic connection of the Lambert function, which is that for any value of $x$ it holds that:

$$W(x) = \ln(x) - ln(W(x))$$

So for $\alpha \geq \frac{|\mathcal{F}|}{|\mathcal{R}|}$, since $\mathrm{W}_0$ is increasing we have that $w_j^{\mathcal{D}} \geq w_j^{\mathcal{R}}$ we have the following:

$$
\begin{aligned}
\Delta_{\mathcal{R},\mathcal{D}} &= w_j^{\mathcal{D}} - w_j^{\mathcal{R}} \\
&= \mathrm{W}_0\left(\frac{(1+\alpha)|\mathcal{R}_j|}{\lambda|\mathcal{D}|}\right) - \mathrm{W}_0\left(\frac{|\mathcal{R}_j|}{\lambda|\mathcal{R}|}\right) \\
&= \mathrm{W}_0\left(\alpha\frac{|\mathcal{R}_j|}{\lambda|\mathcal{R}|}\right) - \mathrm{W}_0\left(\frac{|\mathcal{R}_j|}{\lambda|\mathcal{R}|}\right) \quad \text{,where } \alpha = (1+\alpha)\frac{|\mathcal{R}|}{|\mathcal{D}|} \\
&= \ln\left(\alpha\frac{|\mathcal{R}_j|}{\lambda|\mathcal{R}|}\right) - \ln\left(\mathrm{W}_0\left(\alpha\frac{|\mathcal{R}_j|}{\lambda|\mathcal{R}|}\right)\right) - \ln\left(\frac{|\mathcal{R}_j|}{\lambda|\mathcal{R}|}\right) + \ln\left(\mathrm{W}_0\left(\frac{|\mathcal{R}_j|}{\lambda|\mathcal{R}|}\right)\right) \\
&= \ln\left(\alpha\frac{|\mathcal{R}_j|}{\lambda|\mathcal{R}|}\right) - \ln\left(\mathrm{W}_0\left(\alpha\frac{|\mathcal{R}_j|}{\lambda|\mathcal{R}|}\right)\right) - \ln\left(\frac{|\mathcal{R}_j|}{\lambda|\mathcal{R}|}\right) + \ln\left(\mathrm{W}_0\left(\frac{|\mathcal{R}_j|}{\lambda|\mathcal{R}|}\right)\right) \\
&= \ln\left(\alpha\right) - \ln\left(\frac{\mathrm{W}_0\left(\alpha\frac{|\mathcal{R}_j|}{\lambda|\mathcal{R}|}\right)}{\mathrm{W}_0\left(\frac{|\mathcal{R}_j|}{\lambda|\mathcal{R}|}\right)}\right) \\
&\leq \ln\left(\alpha\right) \quad \text{, since the principal branch } \mathrm{W}_0 \text{ is increasing}
\end{aligned}
$$

We can repeat the same proof procedure for $\alpha \leq \frac{|\mathcal{F}|}{|\mathcal{R}|}$, but instead we get $\Delta_{\mathcal{R},\mathcal{D}} \leq -\ln(\alpha)$. This concludes the proof $\qquad\square$

## H  Proof of Lemma 5

**Lemma 5** (Distance Unlearning). *Let $w_j^{\mathcal{R}}, w_j^{DA}$ the $j^{th}$ coordinate of the convergence point for the logistic regression problem for the retain set $\mathcal{R}$ and the Descent Ascent method respectively. It holds that for for $\alpha \geq |\mathcal{F}|/|\mathcal{R}|$ the distance $\Delta_{\mathcal{R},DA} = |w_j^{\mathcal{R}} - w_j^{DA}| \geq W_0\left(|\mathcal{R}_j|/(\lambda|\mathcal{R}|)\right)$*

*Proof.* We start from Lemma 2 which we restate below for the sake of exposition.

**Lemma 2** (Closed Form). *Let $w_j^{\mathcal{D}}, w_j^{\mathcal{R}}$ and $w_j^{DA}$ be the $j^{th}$ coordinate of **any** local minima/maxima for the logistic regression problems defined in Eq. (1), then they admit the form:*

$$
w_j^{\mathcal{D}} = \mathrm{W}\left(\frac{(1+\alpha)|\mathcal{R}_j|}{\lambda|\mathcal{D}|}\right), w_j^{\mathcal{R}} = \mathrm{W}\left(\frac{|\mathcal{R}_j|}{\lambda|\mathcal{R}|}\right), w_j^{DA} = \mathrm{W}\left(\frac{(1-\alpha|\mathcal{R}|/|\mathcal{F}|)|R_j|}{\lambda|R|}\right),
$$

*where $W(z)$ corresponds to the Lambert-W function, the solution to $z = W(z)e^{W(z)}$.*

It is easy to notice that in the case where we have $\alpha = |\mathcal{F}|/|\mathcal{R}|$ $w_j^{DA} = 0$ which concludes this case. For the case where $\alpha > |\mathcal{F}|/|\mathcal{R}|$ we refer the reader to the proof of Lemma 3, where we show that $w_j^{DA} \to -\infty$ for any value of $\lambda > 0$ so the distance is infinite in this case. $\qquad\square$

## I  Logistic Regression 2 dimensions

In this section we will study the natural extension of the previous example, where we were studying the 1 dimensional case. In this case we assume that our samples are of the form:

$$
s_1 = (1, \epsilon), \quad s_2 = (\epsilon, 1)
$$

This gives the following equations for the optimality conditions for training on the full data set $\mathcal{D}$:

$$
\begin{aligned}
w_1 &= \frac{|\mathcal{R}_{i,j}|}{\lambda|\mathcal{D}|}\left(e^{-(w_1+w_2\epsilon)} + \alpha\epsilon e^{-(w_1\epsilon+w_2)}\right) \\
w_2 &= \frac{|\mathcal{R}_{i,j}|}{\lambda|\mathcal{D}|}\left(\epsilon e^{-(w_1+w_2\epsilon)} + \alpha e^{-(w_1\epsilon+w_2)}\right)
\end{aligned}
$$

For the retrain set $\mathcal{R}$ we have that:

$$w_1 = \frac{|\mathcal{R}_{i,j}|}{\lambda|\mathcal{R}|}e^{-(w_1+w_2\epsilon)}$$

$$w_2 = \frac{|\mathcal{R}_{i,j}|}{\lambda|\mathcal{R}|}\epsilon e^{-(w_1+w_2\epsilon)}$$

For the Descent Ascent unlearning we have that:

$$w_1 = \frac{|\mathcal{R}_{i,j}|}{\lambda|\mathcal{R}|}e^{-(w_1+w_2\epsilon)} - \frac{|\mathcal{R}_{i,j}|}{\lambda|\mathcal{F}|}\alpha\epsilon e^{-(w_1\epsilon+w_2)}$$

$$w_2 = \frac{|\mathcal{R}_{i,j}|}{\lambda|\mathcal{R}|}\epsilon e^{-(w_1+w_2\epsilon)} - \frac{|\mathcal{R}_{i,j}|}{\lambda|\mathcal{F}|}\alpha e^{-(w_1\epsilon+w_2)}$$

We will now rewrite the above equations by setting $x = w_1 + w_2\epsilon$ and $y = w_1\epsilon + w_2$, this simplifies the equations and still allows us to make our claim that DA can only harm the model if there is a total ordering over the values of the solutions of the rewritten equations.

For the dataset $\mathcal{D}$ we have:

$$x^{\mathcal{D}} = \frac{|\mathcal{R}_{i,j}|}{\lambda|\mathcal{D}|}((1+\epsilon^2)e^{-x^{\mathcal{D}}} + 2\alpha\epsilon e^{-y^{\mathcal{D}}})$$

$$y^{\mathcal{D}} = \frac{|\mathcal{R}_{i,j}|}{\lambda|\mathcal{D}|}(2\epsilon e^{-x^{\mathcal{D}}} + \alpha(1+\epsilon^2)e^{-y^{\mathcal{D}}})$$

For the retrain set $\mathcal{R}$, we have that:

$$x^{\mathcal{R}} = \frac{|\mathcal{R}_{i,j}|}{\lambda|\mathcal{R}|}(1+\epsilon^2)e^{-x^{\mathcal{R}}}$$

$$y^{\mathcal{R}} = \frac{|\mathcal{R}_{i,j}|}{\lambda|\mathcal{R}|}2\epsilon e^{-x^{\mathcal{R}}}$$

For the DA method we get the following equations:

$$x^{\mathrm{DA}} = \frac{|\mathcal{R}_{i,j}|}{\lambda|\mathcal{R}|}(1+\epsilon^2)e^{-x^{\mathrm{DA}}} - \frac{|\mathcal{R}_{i,j}|}{\lambda|\mathcal{F}|}\alpha 2\epsilon e^{-y^{\mathrm{DA}}}$$

$$y^{\mathrm{DA}} = \frac{|\mathcal{R}_{i,j}|}{\lambda|\mathcal{R}|}2\epsilon e^{-x^{\mathrm{DA}}} - \frac{|\mathcal{R}_{i,j}|}{\lambda|\mathcal{F}|}\alpha(1+\epsilon^2)e^{-y^{\mathrm{DA}}}$$

Before proceeding, let us point out that $y^{\mathrm{DA}} \leq x^{\mathrm{DA}}$, since $1 + \epsilon^2 \geq 2\epsilon$, for the same reason, we get that $y^{\mathcal{R}} \leq x^{\mathcal{R}}$ and finally without loss of generality we will use that $y^{\mathcal{D}} \leq x^{\mathcal{D}}$. In Lemma 12 we give a short proof regarding the existence of such solutions.

**Lemma 12.** *For any $\alpha \leq 1$, we have that there exists a solution for the original dataset, such that $y^{\mathcal{D}} \leq x^{\mathcal{D}}$*

*Proof.* For $\alpha = 1$ we get that there exists a solution of the system such that $y^{\mathcal{D}} \leq x^{\mathcal{D}}$ by the symmetry of the system. For $\alpha \leq 1$. In order to demonstrate that there exists a solution for the system such that $y^{\mathcal{D}} \leq x^{\mathcal{D}}$ we will employ the nonlinear Gauss-Sidel method, which converges to a stationary point (minimum) for logistic regression. The proof goes as follows, we will initialize our

algorithm in the solution for $\alpha = 1$ let it be $x_0, y_0$ and we know it holds that $x_0 \geq y_0$. We will follow the following update: (nonlinear Gauss-Sidel method starting from $y$)

$$y_{k+1} \leftarrow 2b\epsilon e^{-x_k} + W\left(b\alpha(1+\epsilon^2)e^{-2b\epsilon e^{-x_k}}\right)$$

$$x_{k+1} \leftarrow 2b\alpha\epsilon e^{-y_k} + W\left(b(1+\epsilon^2)e^{-2b\alpha\epsilon e^{-y_k}}\right)$$

For $y_1$ we have that:

$$
\begin{aligned}
y_1 &= 2b\epsilon e^{-x_0} + W\left(b\alpha(1+\epsilon^2)e^{-2b\epsilon e^{-x_0}}\right) \\
&= 2b\epsilon e^{-x_0} + W\left(b\alpha(1+\epsilon^2)e^{-2b\epsilon e^{-x_0}}\right) - W\left(b(1+\epsilon^2)e^{-2b\epsilon e^{-x_0}}\right) + W\left(b(1+\epsilon^2)e^{-2b\epsilon e^{-x_0}}\right) \\
&= y_0 + W\left(b\alpha(1+\epsilon^2)e^{-2b\epsilon e^{-x_0}}\right) - W\left(b(1+\epsilon^2)e^{-2b\epsilon e^{-x_0}}\right)
\end{aligned}
$$

and since W is increasing we have that $W\left(b\alpha(1+\epsilon^2)e^{-2b\epsilon e^{-x_0}}\right) - W\left(b(1+\epsilon^2)e^{-2b\epsilon e^{-x_0}}\right) < 0$ implying that $y_1 < y_0$. Now let us define the function $f(x) = x + W(ce^{-x})$ the function is increasing on $x$ therefore since $y_1 < y_0$ we get that: $x_1 = f(2b\alpha\epsilon e^{-y_1}) > f(2b\alpha\epsilon e^{-y_0}) = x_0$. Let us proceed with an induction step, we assume that we have $x_k > x_{k-1}$ and $y_k < y_{k-1}$ for $k \geq 1$. We will show that $y_{k+1} < y_k$ which directly implies that $x_{k+1} = f(2b\alpha\epsilon e^{-y_{k+1}}) > f(2b\alpha\epsilon e^{-y_k}) = x_k$ completing the inductive step.

$$
\begin{aligned}
y_{k+1} &= 2b\epsilon e^{-x_k} + W\left(b\alpha(1+\epsilon^2)e^{-2b\epsilon e^{-x_k}}\right) \\
&= f(2b\epsilon e^{-x_k}) < f(2b\epsilon e^{-x_{k-1}}) \\
&= y_k
\end{aligned}
$$

This concludes the inductive step and we therefore have that for all $k$ $y_k \leq x_k$ for any $\alpha$, as a result, since the method converges to the solution of the system there exists a solution which satisfies $y^{\mathcal{D}} \leq x^{\mathcal{D}}$. In the proof above we have that $b = ||\mathcal{R}_{i,j}|/\lambda|\mathcal{D}|$ $\qquad\square$

### I.1 Characterization of the solutions of the $2d$ Logistic regression

We start this section by giving an exact solution for the coordinates of the retrain problem.

**Lemma 6.** *The closed form solution for the stationary points for the retrain set is given as:*

$$x^{\mathcal{R}} = W\left(\frac{(1+\epsilon^2)|R_{i,j}|}{\lambda|\mathcal{R}|}\right), \quad y^{\mathcal{R}} = \frac{2\epsilon}{1+\epsilon^2}W\left(\frac{(1+\epsilon^2)|R_{i,j}|}{\lambda|\mathcal{R}|}\right).$$

*Proof.* We have that:

$$x^{\mathcal{R}} = \frac{|\mathcal{R}_{i,j}|}{\lambda|\mathcal{R}|}(1+\epsilon^2)e^{-x^{\mathcal{R}}} \rightarrow x^{\mathcal{R}} = W\left(\frac{(1+\epsilon^2)|R_{i,j}|}{\lambda|\mathcal{R}|}\right)$$

So:

$$
\begin{aligned}
y^{\mathcal{R}} &= \frac{|\mathcal{R}_{i,j}|}{\lambda|\mathcal{R}|}2\epsilon e^{-x^{\mathcal{R}}} \\
&= \frac{2\epsilon}{1+\epsilon^2}\frac{(1+\epsilon^2)|R_{i,j}|}{\lambda|\mathcal{R}|}e^{-W\left(\frac{(1+\epsilon^2)|R_{i,j}|}{\lambda|\mathcal{R}|}\right)} \\
&= \frac{2\epsilon}{1+\epsilon^2}W\left(\frac{(1+\epsilon^2)|R_{i,j}|}{\lambda|\mathcal{R}|}\right)
\end{aligned}
$$

This concludes the proof. In the last equality we used the property of the Lambert function. $\qquad\square$

For the other two problems it is not possible to provide exact solutions, as we did in the retrain one unfortunately, so we will provide upper and lower bounds for their values.

**Lemma 7.** *For the stationary points of the original set, one can derive the following ranges.*

$$W\left(\frac{|R_{i,j}|}{\lambda|\mathcal{D}|}((1+\epsilon^2)+2\alpha\epsilon)\right) \leq x^{\mathcal{D}} \leq \frac{2\epsilon}{1+\epsilon^2}W\left(\frac{\alpha(1+\epsilon^2)|R_{i,j}|}{\lambda|\mathcal{D}|}\right) + W\left(\frac{(1+\epsilon^2)|R_{i,j}|}{\lambda|\mathcal{D}|}\right),$$

$$W\left(\frac{\alpha(1+\epsilon^2)|R_{i,j}|}{\lambda|\mathcal{D}|}\right) \leq y^{\mathcal{D}} \leq W\left(\frac{|R_{i,j}|}{\lambda|\mathcal{D}|}(2\epsilon+\alpha(1+\epsilon^2))\right).$$

*Proof.* We have that

$$x^{\mathcal{D}} = \frac{1}{\lambda|\mathcal{D}|}((1+\epsilon^2)e^{-x^{\mathcal{D}}} + 2\alpha\epsilon e^{-y^{\mathcal{D}}})$$

$$y^{\mathcal{D}} = \frac{1}{\lambda|\mathcal{D}|}(2\epsilon e^{-x^{\mathcal{D}}} + \alpha(1+\epsilon^2)e^{-y^{\mathcal{D}}})$$

As we discuss above we have that $y^{\mathcal{D}} \leq x^{\mathcal{D}} \Rightarrow e^{-y^{\mathcal{D}}} \geq e^{-x^{\mathcal{D}}}$, which implies that:

$$x^{\mathcal{D}} \geq \frac{|\mathcal{R}_{i,j}|}{\lambda|\mathcal{D}|}((1+\epsilon^2)e^{-x^{\mathcal{D}}} + 2\alpha\epsilon e^{-x^{\mathcal{D}}}) = \frac{|\mathcal{R}_{i,j}|}{\lambda|\mathcal{D}|}((1+\epsilon^2)+2\alpha\epsilon)e^{-x^{\mathcal{D}}}$$

$$y^{\mathcal{D}} \leq \frac{|\mathcal{R}_{i,j}|}{\lambda|\mathcal{D}|}(2\epsilon e^{-y^{\mathcal{D}}} + \alpha(1+\epsilon^2)e^{-y^{\mathcal{D}}}) = \frac{|\mathcal{R}_{i,j}|}{\lambda|\mathcal{D}|}(2\epsilon + \alpha(1+\epsilon)^2)e^{-y^{\mathcal{D}}}$$

So from the inequalities above, we get that:

$$x^{\mathcal{D}} \geq W\left(\frac{|\mathcal{R}_{i,j}|}{\lambda|\mathcal{D}|}((1+\epsilon^2)+2\alpha\epsilon)\right)$$

$$y^{\mathcal{D}} \leq W\left(\frac{|\mathcal{R}_{i,j}|}{\lambda|\mathcal{D}|}(2\epsilon+\alpha(1+\epsilon^2))\right)$$

Now we have an upper bound for $y^{\mathcal{D}}$ and a lower bound for $x^{\mathcal{D}}$. In order to provide a lower bound for $y^{\mathcal{D}}$ and an upper bound for $x^{\mathcal{D}}$. We should notice that $2\epsilon e^{-x^{\mathcal{D}}} \geq 0$, which gives:

$$y^{\mathcal{D}} \geq \frac{|\mathcal{R}_{i,j}|}{\lambda|\mathcal{D}|}\alpha(1+\epsilon^2)e^{-y^{\mathcal{D}}} \Rightarrow$$

$$y^{\mathcal{D}} \geq W\left(\frac{\alpha(1+\epsilon^2)|\mathcal{R}_{i,j}|}{\lambda|\mathcal{D}|}\right)$$

This completes the bounds for $y^{\mathcal{D}}$, now in order to compute the upper bound for $x^{\mathcal{D}}$, we have that:

$$e^{-y^{\mathcal{D}}} \leq e^{-W\left(\alpha(1+\epsilon^2)|\mathcal{R}_{i,j}|/(\lambda|\mathcal{D}|)\right)} \Rightarrow$$

$$e^{-y^{\mathcal{D}}} \leq \frac{\lambda|\mathcal{D}|}{\alpha(1+\epsilon^2)|\mathcal{R}_{i,j}|}\frac{\alpha(1+\epsilon^2)|\mathcal{R}_{i,j}|}{\lambda|\mathcal{D}|}e^{-W\left(\alpha(1+\epsilon^2)|\mathcal{R}_{i,j}|/(\lambda|\mathcal{D}|)\right)} \Rightarrow$$

$$e^{-y^{\mathcal{D}}} \leq \frac{\lambda|\mathcal{D}|}{\alpha(1+\epsilon^2)|\mathcal{R}_{i,j}|}W\left(\frac{\alpha(1+\epsilon^2)|\mathcal{R}_{i,j}|}{\lambda|\mathcal{D}|}\right)$$

So we have that:

$$x^{\mathcal{D}} \leq \frac{|\mathcal{R}_{i,j}|}{\lambda|\mathcal{D}|}((1+\epsilon^2)e^{-x^{\mathcal{D}}} + 2\alpha\epsilon\frac{\lambda|\mathcal{D}|}{\alpha(1+\epsilon^2)|\mathcal{R}_{i,j}|}W\left(\frac{\alpha(1+\epsilon^2)}{\lambda|\mathcal{D}|}\right)) \Rightarrow$$

$$x^{\mathcal{D}} \leq \frac{(1+\epsilon^2)|\mathcal{R}_{i,j}|}{\lambda|\mathcal{D}|}e^{-x^{\mathcal{D}}} + \frac{2\epsilon}{1+\epsilon^2}W\left(\frac{\alpha(1+\epsilon^2)|\mathcal{R}_{i,j}|}{\lambda|\mathcal{D}|}\right)) \Rightarrow$$

$$x^{\mathcal{D}} \leq \frac{2\epsilon}{1+\epsilon^2}W\left(\frac{\alpha(1+\epsilon^2)|\mathcal{R}_{i,j}|}{\lambda|\mathcal{D}|}\right) + W\left(\frac{(1+\epsilon^2)|\mathcal{R}_{i,j}|}{\lambda|\mathcal{D}|}e^{-\frac{2\epsilon}{1+\epsilon^2}W\left(\frac{\alpha(1+\epsilon^2)|\mathcal{R}_{i,j}|}{\lambda|\mathcal{D}|}\right)}\right)) \Rightarrow$$

$$x^{\mathcal{D}} \leq \frac{2\epsilon}{1+\epsilon^2}W\left(\frac{\alpha(1+\epsilon^2)|\mathcal{R}_{i,j}|}{\lambda|\mathcal{D}|}\right) + W\left(\frac{(1+\epsilon^2)|\mathcal{R}_{i,j}|}{\lambda|\mathcal{D}|}\right)$$

where the third inequality comes from the solution of the Lambert equation for the RHS of the inequality and the last one comes from the fact that the exponent is non positive. This completes the proof. ☐

**Lemma 8.** *For the stationary points of the model trained by DA methods we can derive the following ranges.*

$$x^{DA} \leq W\left(\frac{|R_{i,j}|}{\lambda|\mathcal{R}|}(1+\epsilon^2) - \frac{|R_{i,j}|}{\lambda|\mathcal{F}|}\alpha 2\epsilon\right), \qquad y^{DA} \leq W\left(\frac{|R_{i,j}|}{\lambda|\mathcal{R}|}2\epsilon - \frac{|R_{i,j}|}{\lambda|\mathcal{F}|}\alpha(1+\epsilon^2)\right).$$

*Proof.* As stated earlier we have that $y^{\mathrm{DA}} \leq x^{\mathrm{DA}} \Rightarrow e^{-y^{\mathrm{DA}}} \geq e^{-x^{\mathrm{DA}}}$ and

$$x^{\mathrm{DA}} = \frac{|\mathcal{R}_{i,j}|}{\lambda|\mathcal{R}|}(1+\epsilon^2)e^{-x^{\mathrm{DA}}} - \frac{|\mathcal{R}_{i,j}|}{\lambda|\mathcal{F}|}\alpha 2\epsilon e^{-y^{\mathrm{DA}}}$$

$$y^{\mathrm{DA}} = \frac{|\mathcal{R}_{i,j}|}{\lambda|\mathcal{R}|}2\epsilon e^{-x^{\mathrm{DA}}} - \frac{|\mathcal{R}_{i,j}|}{\lambda|\mathcal{F}|}\alpha(1+\epsilon^2)e^{-y^{\mathrm{DA}}}$$

So:

$$x^{\mathrm{DA}} \leq \left(\frac{|\mathcal{R}_{i,j}|}{\lambda|\mathcal{R}|}(1+\epsilon^2) - \frac{|\mathcal{R}_{i,j}|}{\lambda|\mathcal{F}|}\alpha 2\epsilon\right)e^{-x^{\mathrm{DA}}}$$

$$y^{\mathrm{DA}} \leq \left(\frac{|\mathcal{R}_{i,j}|}{\lambda|\mathcal{R}|}2\epsilon - \frac{|\mathcal{R}_{i,j}|}{\lambda|\mathcal{F}|}\alpha(1+\epsilon^2)\right)e^{-y^{\mathrm{DA}}}$$

So we get that:

$$x^{\mathrm{DA}} \leq W\left(\frac{|\mathcal{R}_{i,j}|}{\lambda|\mathcal{R}|}(1+\epsilon^2) - \frac{|\mathcal{R}_{i,j}|}{\lambda|\mathcal{F}|}\alpha 2\epsilon\right)$$

$$y^{\mathrm{DA}} \leq W\left(\frac{|\mathcal{R}_{i,j}|}{\lambda|\mathcal{R}|}2\epsilon - \frac{|\mathcal{R}_{i,j}|}{\lambda|\mathcal{F}|}\alpha(1+\epsilon^2)\right)$$

which concludes the proof. $\qquad\square$

## I.2 Derivation of the relevant size of the forget set

**Lemma 9.** *For $\alpha \geq \alpha^{\mathcal{D}>DA} = \max\left\{\frac{1+\epsilon^2}{2\epsilon}\frac{|\mathcal{F}|^2}{|\mathcal{R}|(|\mathcal{D}|+|\mathcal{F}|)}, \frac{2\epsilon}{1+\epsilon^2}\frac{|\mathcal{F}||\mathcal{D}|}{|\mathcal{R}|(|\mathcal{D}|+|\mathcal{F}|)}\right\}$ we have that $x^{\mathcal{D}} \geq x^{DA}$ and that $y^{\mathcal{D}} \geq y^{DA}$.*

*Proof.* We will start from Lemma 7 and Lemma 8, which we restate both below for the sake of exposition.

**Lemma 7.** *For the stationary points of the original set, one can derive the following ranges.*

$$W\left(\frac{|R_{i,j}|}{\lambda|\mathcal{D}|}((1+\epsilon^2)+2\alpha\epsilon)\right) \leq x^{\mathcal{D}} \leq \frac{2\epsilon}{1+\epsilon^2}W\left(\frac{\alpha(1+\epsilon^2)|R_{i,j}|}{\lambda|\mathcal{D}|}\right) + W\left(\frac{(1+\epsilon^2)|R_{i,j}|}{\lambda|\mathcal{D}|}\right),$$

$$W\left(\frac{\alpha(1+\epsilon^2)|R_{i,j}|}{\lambda|\mathcal{D}|}\right) \leq y^{\mathcal{D}} \leq W\left(\frac{|R_{i,j}|}{\lambda|\mathcal{D}|}(2\epsilon + \alpha(1+\epsilon^2))\right).$$

**Lemma 8.** *For the stationary points of the model trained by DA methods we can derive the following ranges.*

$$x^{DA} \leq W\left(\frac{|R_{i,j}|}{\lambda|\mathcal{R}|}(1+\epsilon^2) - \frac{|R_{i,j}|}{\lambda|\mathcal{F}|}\alpha 2\epsilon\right), \qquad y^{DA} \leq W\left(\frac{|R_{i,j}|}{\lambda|\mathcal{R}|}2\epsilon - \frac{|R_{i,j}|}{\lambda|\mathcal{F}|}\alpha(1+\epsilon^2)\right).$$

We will require that the lower bounds provided for $x^{\mathcal{D}}, y^{\mathcal{D}}$ are bigger than the upper bounds provided for $x^{\mathrm{DA}}, y^{\mathrm{DA}}$, since the Lambert function W is monotone, we can just solve both inequalities for $\alpha$, $x^{\mathcal{D}} \geq x^{\mathrm{DA}}$ and $y^{\mathcal{D}} \geq y^{\mathrm{DA}}$ and this concludes the proof. $\qquad\square$

Finally we need to find the range of $\alpha$ for which it holds that $x^{\mathcal{R}} \geq x^{\mathcal{D}}$ and $y^{\mathcal{R}} \geq y^{\mathcal{D}}$, which is given in Lemma 10, which we restate next for the sake of exposition.

**Lemma 10.** *For $\alpha \leq \alpha^{\mathcal{R}>\mathcal{D}} = \min\left\{\alpha_x^{\mathcal{R}>\mathcal{D}}, \alpha_y^{\mathcal{R}>\mathcal{D}}\right\}$ we have that $x^{\mathcal{R}} \geq x^{\mathcal{D}}$ and that $y^{\mathcal{R}} \geq y^{\mathcal{D}}$, with $\alpha_x^{\mathcal{R}>\mathcal{D}}, \alpha_y^{\mathcal{R}>\mathcal{D}}$.*

*Proof.* We will use  Lemma 7 and  Lemma 6. Again similar to  Lemma 9 we can solve for $\alpha$ and we get the expressions that solve the $x^\mathcal{R} > x^\mathcal{D}, y^\mathcal{R} > y^\mathcal{D}$ equations. Solving $x^\mathcal{R} = x^\mathcal{D}$

$$\alpha_x^{\mathcal{R} > \mathcal{D}} = \frac{D\lambda \left( W\left(\frac{\epsilon^2+1}{\lambda R}\right) - W\left(\frac{\epsilon^2+1}{D\lambda}\right) \right) \exp\left( \frac{(\epsilon^2+1)\left(W\left(\frac{\epsilon^2+1}{\lambda R}\right) - W\left(\frac{\epsilon^2+1}{D\lambda}\right)\right)}{2\epsilon} \right)}{2\epsilon}, \tag{6}$$

where for any $\alpha < \alpha_x^{\mathcal{R} > \mathcal{D}}$ there is a range of $\epsilon$ for which $x^\mathcal{R} > x^\mathcal{D}$.

Similarly, solving $y^\mathcal{R} = y^\mathcal{D}$

$$\alpha_y^{\mathcal{R} > \mathcal{D}} = \frac{2\epsilon \left( D\lambda e^{\frac{2\epsilon W\left(\frac{\epsilon^2+1}{\lambda R}\right)}{\epsilon^2+1}} W\left(\frac{\epsilon^2+1}{\lambda R}\right) - \epsilon^2 - 1 \right)}{(\epsilon^2+1)^2}, \tag{7}$$

where for any $\alpha < \alpha_y^{\mathcal{R} > \mathcal{D}}$ there is a range of $\epsilon$ for which $y^\mathcal{R} > y^\mathcal{D}$.

The solution is therefore $\alpha \leq \min\left[\alpha_x^{\mathcal{R} > \mathcal{D}}, \alpha_y^{\mathcal{R} > \mathcal{D}}\right]$. $\qquad\square$

## J   Logistic Regression in 2D intuition

Let us consider nearly orthogonal data, such that all coordinates apart from two are orthogonal to each other. Namely, we choose the first two samples to be $x_1 = (1, \epsilon, 0, \ldots, 0)$ and $x_2 = (\epsilon, 1, 0, \ldots, 0)$, while the remaining $d - 2$ points are orthogonal such that $x_a = e_a$ for $a = 3, \ldots, d$, where $e_a$ are the unit vectors. We further assume that the two correlated samples $x_1, x_2$ share the same label $y_1 = y_2 = 1$. In this case, the unlearning problem decouples the first 2 dimensions from the rest, leaving a coupled set of equations for the weights along the first two directions $w_1, w_2$ for the original classification problem

$$w_1 = \frac{1}{\lambda|\mathcal{D}|}(e^{-(w_1+w_2\epsilon)} + \epsilon e^{-(w_1\epsilon+w_2)}), \quad w_2 = \frac{1}{\lambda|\mathcal{D}|}(\epsilon e^{-(w_1+w_2\epsilon)} + e^{-(w_1\epsilon+w_2)}), \tag{8}$$

which can be solved in the limit of $\epsilon \to 1^-$, as

$$w_1 = w_2 = \frac{1}{2}W\left(\frac{2(\epsilon+1)}{\lambda|\mathcal{D}|}\right). \tag{9}$$

The retrain problem has the minimum at

$$w_1 = \frac{1}{\lambda|\mathcal{R}|}e^{-(w_1+w_2\epsilon)}, \quad w_2 = \frac{1}{\lambda|\mathcal{R}|}\epsilon e^{-(w_1+w_2\epsilon)}, \tag{10}$$

and the DA is given by

$$w_1 = \frac{1}{\lambda|\mathcal{R}|}(e^{-(w_1+w_2\epsilon)} - \epsilon e^{-(w_1\epsilon+w_2)}), \quad w_2 = \frac{1}{\lambda|\mathcal{R}|}(\epsilon e^{-(w_1+w_2\epsilon)} - e^{-(w_1\epsilon+w_2)}). \tag{11}$$

Our goal is to study how far is the solution given by GDA from the one given by retraining. The retrained solution can be found analytically to be

$$w_1 = \frac{W\left(\frac{\epsilon^2+1}{|\mathcal{R}|\lambda}\right)}{\epsilon^2+1}, \quad w_2 = \frac{\epsilon W\left(\frac{\epsilon^2+1}{|\mathcal{R}|\lambda}\right)}{\epsilon^2+1}. \tag{12}$$

The GDA equations do not obtain a closed form solution, but they can be solved when assuming $\epsilon \to 1^-$, such that

$$w_1 = \frac{e^{-w_1-w_2}(w_1 - w_2 - 1)(\epsilon - 1)}{\lambda|\mathcal{R}|}, \quad w_2 = \frac{e^{-w_1-w_2}(w_1 - w_2 + 1)(\epsilon - 1)}{\lambda|\mathcal{R}|} \tag{13}$$

which are solved as

$$w_1 = \frac{1}{4}\left( W\left(-\frac{8(\epsilon-1)^2}{|\mathcal{R}|^2\lambda^2}\right) - i\sqrt{2}\sqrt{W\left(-\frac{8(\epsilon-1)^2}{|\mathcal{R}|^2\lambda^2}\right)} \right), \tag{14}$$

$$w_2 = \frac{1}{4}\left( W\left(-\frac{8(\epsilon-1)^2}{|\mathcal{R}|^2\lambda^2}\right) + i\sqrt{2}\sqrt{W\left(-\frac{8(\epsilon-1)^2}{|\mathcal{R}|^2\lambda^2}\right)} \right).$$

It is sufficiently interesting to consider the sum of $w_1 + w_2$ compared to the retrained solution, and define the difference

$$\Delta = w_1^{\text{DA}} + w_2^{\text{DA}} - (w_1^{\text{Re}} + w_2^{\text{Re}}) = \frac{1}{2}W\left(-\frac{8(\epsilon-1)^2}{|\mathcal{R}|^2\lambda^2}\right) - \frac{(1+\epsilon)W\left(\frac{\epsilon^2+1}{|\mathcal{R}|\lambda}\right)}{\epsilon^2+1} \qquad (15)$$

$$\underset{\epsilon\to1^-}{=} -W\left(\frac{2}{|\mathcal{R}|\lambda}\right)$$

## K  Experimental details

**Hyperparameters**  Our implementation is based on Rinberg et al. [47] which follows the methodology in Georgiev et al. [38]. We pretrain ResNet-9 for 24 epochs using stochastic gradient descent (SGD) with an initial learning rate of $0.4$, following a cyclic schedule that peaks at epoch 5. We employ a batch size of $512$, momentum of $0.9$, and a weight-decay coefficient of $5 \times 10^{-4}$.

We also adopt nine forget sets directly from Georgiev et al. [38], which comprise both random subsets and semantically coherent subpopulations identified via principal-component analysis of the datamodel influence matrix. To construct them, an $n \times n$ datamodel matrix is formed by concatenating "train×train" datamodels (with $n = 50\,000$) by computing its top principal components (PCs) then we can define:

1. **Forget set 1**: 10 random samples.
2. **Forget set 2**: 100 random samples.
3. **Forget set 3**: 500 random samples.
4. **Forget set 4**: 10 samples with the highest projection onto the 1st PC.
5. **Forget set 5**: 100 samples with the highest projection onto the 1st PC.
6. **Forget set 6**: 250 samples with the highest and 250 samples with the lowest projection onto the 1st PC.
7. **Forget set 7**: 10 samples with the highest projection onto the 2nd PC.
8. **Forget set 8**: 100 samples with the highest projection onto the 2nd PC.
9. **Forget set 9**: 250 samples with the highest and 250 samples with the lowest projection onto the 2nd PC.

Most unlearning algorithms are highly sensitive to the choice of forget set and hyperparameters. Therefore we perform an extensive hyperparameter exploration, evaluating each baseline unlearning algorithm on each forget set. Our setting is again similar to Georgiev et al. [38] but we consider a slightly larger hyperparameter grid for the employed methods and report results for all configurations rather than only the best-performing runs. More specifically, we evaluate over the Cartesian product of the following hyperparameter grids:

- **Gradient Ascent**: Optimized with SGD. Learning rates: $\{1\times10^{-5}, 5\times10^{-5}, 1\times10^{-4}, 5\times10^{-4}, 1\times10^{-3}, 1\times10^{-2}, 5\times10^{-2}\}$; epochs: $\{1, 3, 5, 7, 10\}$.
- **Gradient Descent/Ascent**: Optimized with SGD. Learning rates: $\{5\times10^{-5}, 5\times10^{-4}, 1\times10^{-3}, 5\times10^{-3}\}$; total epochs: $\{5, 7, 10\}$; ascent epochs: $\{3, 5\}$; forget batch size: $\{32, 64\}$.
- **SCRUB**: Optimized with SGD. Learning rates: $\{5\times10^{-5}, 5\times10^{-4}, 1\times10^{-3}, 5\times10^{-3}\}$; total epochs: $\{5, 7, 10\}$; ascent epochs: $\{3, 5\}$; forget batch size: $\{32, 64\}$.

We use a fixed batch size of $64$ and train 100 models per configuration. For each run, we measure performance using the 95-th percentile of KLoM scores.

**Statistical Significance**  Using $N = 100$ models to compute KLoM is computationally expensive although such expense comes at the gain of having low variance and results closely reproducing Georgiev et al. [38]. We find using lower values such as $N = 20$, $N = 50$ to produce large differences between margin distributions of pretrained and oracle models on the retain and validation sets (where KLoM should be low). More specifically, margin distributions become stable for all sets after $N = 80$.

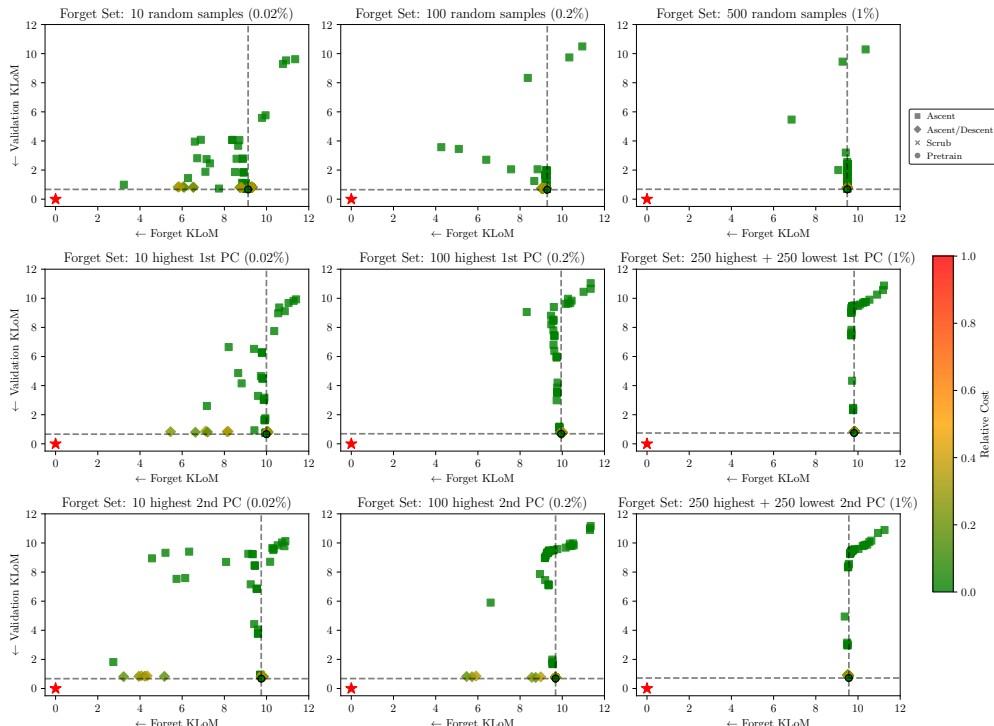

Figure 6: We present the KLoM scores of Gradient Ascent, Gradient Descent/Ascent and SCRUB when unlearning over each one of the forget sets (axes and points follow Fig. 1). We find an increase in forget set size and containing high influence points to strongly decrease the likelihood of any run achieving successful unlearning. For SCRUB we observe that runs remain close to the pretrained model in terms of KLoM scores under our experimental setup.

Reporting the 95-th percentile of KLoM scores follows the methodology established on Georgiev et al. [38]. Furthermore, reporting all runs instead of just the best one for each compute cost is more statistically transparent.

**Compute resources**   All experiments were conducted on a server equipped with eight NVIDIA A100-SXM4 GPUs, each with 80 GB of GPU memory. A single unlearning configuration run was never split across different GPUs, many configurations were executed in parallel.

## L   Additional Experiments

We provide additional analysis of the KLoM scores across various unlearning methods and forget sets. Fig. 6 presents the KLoM scores of Gradient Ascent, Gradient Descent/Ascent, and SCRUB. We observe that increasing the size of the forget set or including high-influence points significantly reduces the likelihood of achieving successful unlearning. Fig. 7 shows analogous results, but with KLoM scores computed over the retain set instead of the validation set. The patterns are nearly identical to those in Fig. 6. A pretrained model typically exhibits low KLoM scores on both validation and retain sets, with very similar magnitudes.

## M   Broader Societal Impact

Machine unlearning is crucial for privacy applications, namely, protecting sensitive data and complying with GDPR's 'right to be forgotten'. Our work, although mainly theoretical, demonstrates that descent-ascent methods often fail due to unacknowledged statistical dependencies between forget and retain sets. This finding has a critical consequence for privacy: to improve ascent based methods, practitioners are required to probe the retain set to understand its correlations with the forget set. This re-assessment of potentially sensitive data in the retain set during an unlearning task creates a privacy

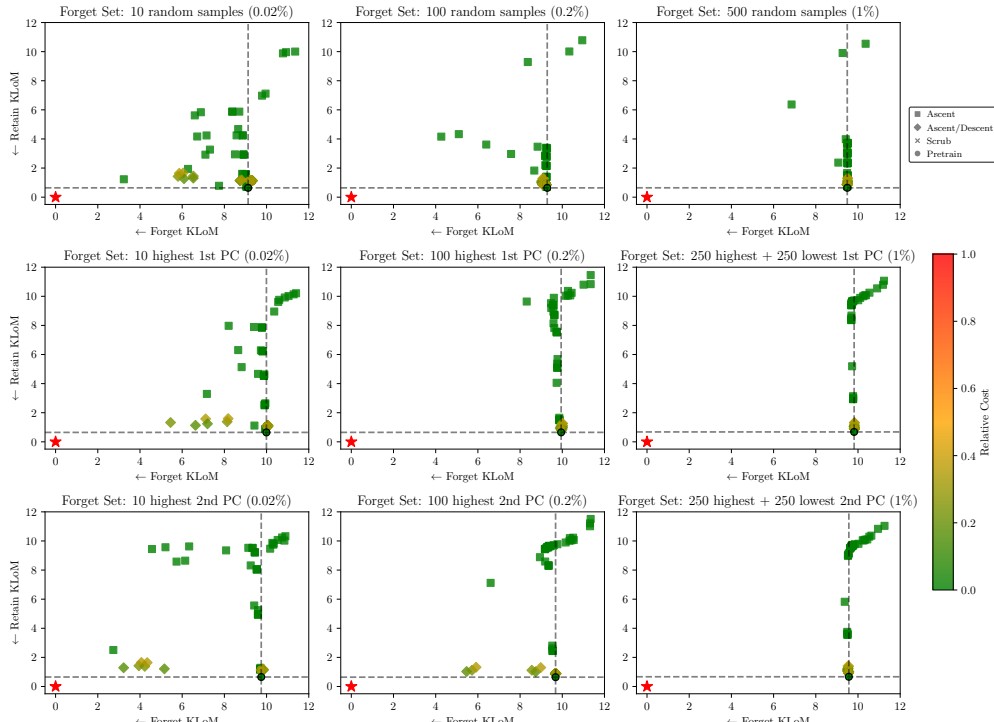

Figure 7: We present the KLoM scores of Gradient Ascent, Gradient Descent/Ascent and SCRUB when unlearning over each forget set. x-axis and points follow Fig. 1 and y-axis now displays the KLoM score in the retain set instead of the validation set. We observe very little difference when comparing with the results in Fig. 6. A pretrained model has low KLoM scores on both the validation and retain sets with very similar magnitudes. These findings are consistent with Georgiev et al. [38].

paradox. Therefore, for applications strictly governed by privacy, alternative unlearning strategies that do not require such re-examination of retained data appear preferable.

