# OpenReview forum: "Ascent Fails to Forget"
_NeurIPS.cc/2025/Conference — NeurIPS 2025 poster_

### Official Review · Reviewer_MLpH · 2025-06-22

**Clarity:** 3
**Significance:** 3
**Originality:** 4
**Rating:** 5
**Confidence:** 5

**Summary:**

This paper critically examines the Descent-Ascent (DA) framework for machine unlearning---a widely adopted method where gradient ascent is applied to forget a subset of training data, followed by gradient descent on the retained set. The authors argue, theoretically, that this method can fundamentally fail to forget. In particular, they show that when there is statistical dependence between forget and retain sets (even if the forget set is selected randomly), DA can cause the model to converge to poor optima and generalize worse than an oracle model retrained without the forget set.

In numerical results, the authors numerically show that the DA-based unlearning degrades the accuracy compared to the oracle model.

**Questions:**

Thank you for your insightful work. I have a few questions and suggestions that I believe could further enhance the clarity and impact of your paper:

1. In Lemma 1, the authors demonstrate that unlearning via distribution alignment (DA) can be detrimental when the accuracies of the test set and forget set are similar. My understanding is that this phenomenon might also lead to degradation in other unlearning methods, such as influence function-based approaches. Could you elaborate on whether this observed issue is a specific limitation of DA-based unlearning, or if it has broader implications for other unlearning paradigms? Furthermore, do you believe certified unlearning methods are robust to the types of failures the authors describe?
2. The authors highlight the importance of KLoM. To further validate their critique, it would be beneficial to see how other exact unlearning methods, such as influence function-based unlearning or retraining approximations, perform in terms of achieving low KLoM. Including such baselines in their analysis could provide a more comprehensive picture.
3. In their numerical results (e.g., Figures 6 and 7), DA-based unlearning often successfully removes the forgotten dataset. If the dataset exhibits extremely high correlation (e.g., MNIST), do DA-based methods consistently fail to unlearn? Could the authors provide further insights into their performance under such highly correlated data conditions?
4. By following the proof of Lemma 1, $ACC_\tau$ denotes the accuracy for the ground truth distribution. Is it normal to represent this accuracy as the test dataset accuracy?
5. KLoM appears to be a metric applicable across various datasets (e.g., MNIST, CIFAR-10). Could the authors explain their rationale for conducting experiments with ImageNet-Living-17 specifically, rather than or in addition to these more common datasets?


Thank you for considering these points. I look forward to the authors' insights.

**Ethical Concerns:**

["NO or VERY MINOR ethics concerns only"]

**Final Justification:**

The authors responded to my comments.
- Lemma 1: All the points are clearly addressed.
- Theoretical expansion: The authors will add directions for future work.
- Explanation of the 1st and 2nd PCs: The authors clearly explain the details.


Although this is an insightful work, I am unable to identify a clear justification for this work to be considered groundbreaking.

**Limitations:**

Yes

**Paper Formatting Concerns:**

No formatting issues

**Quality:**

4

**Strengths And Weaknesses:**

### Strongness

1. **Theoretical Analysis:** It has already been shown that DA approaches can be highly unreliable. However, this work is the first work provide a rigorous mathematical analysis showing the failure of DA even in simple settings, such as logistic regression with correlated inputs. Especially, the analysis on low/high dimensions and cross-domain correlations is impressive.

2. **Timeliness:** There have been several recent studies on DA-based unlearning. With this trend, this work sounds an alarm for requiring alternative approaches other than DA-based unlearning, like rewinding-based methods.

### Weaknesses

1. **Regarding Lemma 1:** While Lemma 1 is presented to support the accuracy of the classification task, the core focus of this work appears to be on regression. This discrepancy can be a bit confusing for the reader. Perhaps the authors could clarify the connection between Lemma 1 and the regression tasks, or consider if its inclusion is essential given the primary focus.
2. **Potential for Theoretical Expansion:** This is an excellent paper as is. However, I believe its impact could be further enhanced by exploring the theoretical extension to multi-layer architectures. This could provide a deeper understanding and broader applicability of this work.
3. **Clarification on Unlearning Performance (Figure 6):** The authors mentioned that training with random samples degrades unlearning performance due to dependency. In Figure 6, the performance degradation is even more pronounced when the first and second principal components (1st/2nd PCs) are set as the forget set, compared to random samples. Could you please elaborate on why this particular choice of forget set leads to a more significant drop in performance?
4. **Minor Suggestion:** In Line 94, the paragraph naming seems inconsistent with the other paragraphs. It might be better to standardize the naming convention for clarity.

I hope these suggestions are helpful for improving this manuscript.

---

> ### Author Rebuttal · Authors · 2025-07-31
>
> We would like to thank the reviewer for their detailed review and the positive evaluation of our work.
>
> Below, we address the specific questions and weaknesses raised.
>
> ### Weaknesses:
>
> - **Regarding Lemma 1:**
> The choice of a classification task in Lemma 1 is arbitrary, and the result can be extended to regression by simply modifying the metric. Accuracy is used solely for clarity and simplicity in the lemma’s formulation. Regarding the connection between regression and classification: logistic regression is a classic example of a regression-based method commonly used for **classification** both in traditional machine learning and in deep learning (in its nonlinear realization), hence we opt for classification throughout this work.
>
> - **Potential for Theoretical Expansion:**
> We agree that extending the result to multi-layer networks would be extremely interesting and broaden the impact of this study. We leave this open for future work.
>
> - **Explaining the 1st and 2nd PC relation to difficulty in unlearning:**
> We believe some intuition can be given with a relatively simple example: Consider once again a 2-dimensional logistic regression task, where labels are generated by a linear transformation of the data $y_i =T  x_i,i=1\ldots N$, where $T\in \mathbb{R}^2$ is a fixed teacher vector. In particular, we can simply choose $T=$ { $1,-1$ } , which would then correspond to classifying points above and below the line $x_{i,1}-x_{i,2}=0$ as negative and positive, respectively. Next, we train a student with weights $S\in \mathbb{R}^2$ to perform logistic regression with labels given by the teacher $T$. If it succeeds, then we expect $S=T$. If we sample points uniformly in a small region near the origin, the influence matrix between a specific point and a test point can be approximated by $I_{z,z_{test}}\approx -y_{test}y \sigma(-y_{test}S x_{test})\sigma(-y S x)x_{test}^T x$ $=-(x_{test,1}-x_{test,2})(x_1-x_2) \sigma(-(x_{test,1}-x_{test,2})^2)\sigma(-(x_1-x_2)^2)x_{test}^T x$, where the inverse Hessian term which should appear is proportional to the identity matrix. If we focus on points scattered on a line of roughly constant $x_1-x_2\approx x_{test,1}-x_{test,2} \approx c$, the amount of correlation between $x_{test}$ and $x$ determines the influence of the point, thus the PCs of $x_{test}^T x$ are expected to align with the influence PCs. We stress that this explanation is only meant to give intuition, and it would be interesting to perform this analysis directly on the full influence matrices in some tractable, as well as realistic settings.
>
> - **Minor Suggestion:**
> We thank the reviewer for pointing out the inconsistency. We made the paragraph style consistent with the paragraph above for the rest of the related work section.
>
> ### Questions:
>
> - **Degradation in other unlearning methods:**
> This is a very interesting question, indeed the findings might be applicable to other schemes. The main intuition is that, when unlearning a random forget set, any method that attempts to approximate an oracle model should have a similar accuracy on the forget set and on the test set, since they are perceived by the network as statistically identical. Consequently, methods which aim to produce a large drop in forget set accuracy will deviate from oracle models. Our findings show that this is indeed a prevalent problem in DA based methods. We do not expect the same to occur in influence function-based approaches and certified-unlearning methods since they do not necessarily target a drop in forget set accuracy.
>
> - **KLoM and baselines:**
> In [1] the authors show how an influence function oriented method (using datamodels) achieves low KLoM scores on the forget set. Since this method does not perform gradient ascent we left it out of this work. We agree it is important to explicitly mention ``datamodels'' success and we will include it in our discussion of related methods.
>
> - **MNIST and high correlations:**
>     Thank you for this insightful question about DA-based methods on highly correlated datasets like MNIST.
>
>     We argue that **MNIST is fundamentally degenerate for machine unlearning evaluation**, making it difficult to assess whether any unlearning method (including DA-based approaches) has truly succeeded. The core issue is that most MNIST points are highly similar to multiple other points in the dataset. When we retrain an oracle model that has never seen the forget set points, its distribution of margins remains very similar to the pretrained model, even on the forget set, making it hard to distinguish between the two.
>
>     To demonstrate this degeneracy, we compare pretrained models (trained on the full dataset) with oracle models (trained without forget sets, requiring no unlearning whatsoever). Previously, on **CIFAR-10**, oracle and pretrained models show large representational differences on forget sets (high 95th percentile KLoM scores) while remaining similar on retained data, consistent with [1]. This pattern indicates that forget set points were meaningfully learned and their absence creates detectable differences.
>
>     **MNIST exhibits the opposite pattern:** Oracle and pretrained models remain nearly identical even on forget sets, with consistently low KLoM scores across all data splits. This similarity persists even when scaling forget set size to 10% of training data. We also tested on **FashionMNIST** which shows some increase in forget set KLoM scores, though not approaching CIFAR-10 magnitudes and clear distinguishability from pretrained models.
>
>     This fundamental issue means that any unlearning method - whether DA-based or otherwise - could appear to "succeed" on MNIST simply by "doing nothing" (e.g. very small learning rate or number of steps) because the dataset's inherent redundancy makes meaningful forgetting hard to verify.
>
>     |Dataset|Forget %|Forget KLoM (95th)|Retain KLoM (95th)|Val KLoM (95th)|
>     |---|---|---|---|---|
>     |MNIST|0.02%|0.5|0.7|0.71|
>     |MNIST|0.2%|2.72|2.88|2.9|
>     |MNIST|10%|1.79|1.65|1.71|
>     |FashionMNIST|0.02%|4.13|1.87|2.76|
>     |FashionMNIST|0.2%|2.72|1.79|2.70|
>     |FashionMNIST|10%|2.69|2.0|3.31|
>
>     For reference, we trained 100 pretrained models for each dataset and 100 oracles for each forget set as in our original experiments. We will add this insights to the appendix for the camera ready version.
>
> - **Accuracy of test:**
> Yes, it is standard practice to represent the ground truth distribution accuracy using test dataset accuracy. The test set serves as an unbiased estimator of the true population performance, as it remains unseen during model development. By the law of large numbers, this estimator converges to the true accuracy as the test set size increases. Since the ground truth distribution is inherently unknown in practice, the test set accuracy provides the most principled approximation available. This approach aligns with established machine learning evaluation protocols where test performance serves as the gold standard for assessing generalization capability.
>
> - **ImageNet-Living-17:**
> We apologize for this it was a typo we meant to write cifar-10 here.
>
> [1] Georgiev et al "Attribute-to-Delete: Machine Unlearning via Datamodel Matching." arXiv:2410.23232v2

---

> > ### Comment · Reviewer_MLpH · 2025-08-02
> >
> > To the authors,
> >
> > Thank you for your time in writing the responses to my comments, especially for additional experiments on the MNIST datasets. I thoroughly read the point-by-point responses.
> > - The reviewer agrees that the DA can be degraded if datasets are stochastically identical.
> > - **Additional Comment:** For the below response, the estimator converges to the true accuracy as **test set size increases.** So, doesn't that require additional assumptions for your theoretical results?
> > >Yes, it is standard practice to represent the ground truth distribution accuracy using test dataset accuracy. The test set serves as an unbiased estimator of the true population performance, as it remains unseen during model development. By the law of large numbers, this estimator converges to the true accuracy as the test set size increases.
> >
> > Once again, sincerely appreciate the detailed and kind responses.

---

> ### Author Response · Authors · 2025-08-03
>
> To the reviewer,
>
> Thank you for your quick reply. Let us provide some additional clarification, regarding the additional comment.
>
> In practice, the goal is to deploy a model that performs optimally under the ground truth distribution, as this would yield the best expected performance on an unseen test set, whose exact composition is not known in advance. As an additional clarification, we noted that test performance converges to the expected performance under the ground truth distribution as the test set size grows. Ultimately, the primary objective is to achieve the highest possible accuracy with respect to the ground truth distribution, thereby ensuring that the model generalizes well to unseen test data, consequently, we don't require additional assumptions, for our theoretical results.

---

> > ### Comment · Reviewer_MLpH · 2025-08-08
> >
> > Thank you for your clarification on the additional comments.
> > I will maintain my positive score (5: acceptance). Although this is an insightful work, I am unable to identify a clear justification for this work to be considered groundbreaking.
> > If you have an additional response to this comment, please let me know.
> >
> > I sincerely sorry for the lateness of this response.

---

### Official Review · Reviewer_85tp · 2025-06-28

**Clarity:** 2
**Significance:** 3
**Originality:** 3
**Rating:** 4
**Confidence:** 4

**Summary:**

The paper focuses on understanding the weaknesses of gradient ascent based methods in the context of machine unlearning.

First, it presents experiments on CIFAR-10, showing that gradient ascent based methods either do not move from the initial model parameters or severely degrade model performance. Moreover, their performance is fragile and sensitive to choice of hyperparameters.

Then, the paper argues that presence of correlations between retain set points and forget set points causes ascent based methods to degrade the model performance and not do unlearning. The paper presents this as a series of claims:

- First, using standard concentration bounds, it shows that if the forget set is a random subset of training set, then the average accuracy of a model on training points and forget points can't be very different. Using this result, it argues that ascent based methods that actively try to decrease forget set accuracy must harm model performance.

- Second, it considers some stylized settings where there is correlation between retain and forget points, where unlearning using ascent based methods is worse than doing nothing.

- Finally, it shows another 2-dimensional example, where even if you use gradient descent  on retain set after using gradient ascent, gradient descent can not undo the harm done by gradient ascent.

**Questions:**

The paper needs significant rewriting for clarity, along with significantly more convincing and thorough demonstration of the claims. Please refer to the strengths and weaknesses section for more details ans questions.

**Ethical Concerns:**

["NO or VERY MINOR ethics concerns only"]

**Final Justification:**

I thank the authors for the detailed response. Most of my questions on the relevance of the analysis are answered. It would be great to add some of the discussion in the response to the next version of the paper, and also improve the clarity of the writing.

The part that still seems a bit lacking is the empirical demonstartion in that there is not enough convincing demonstration that correlations are a major reason behind the faliure of gradient ascent based methods in practice. Also, the precise definition of correlation when we move away from stylized settings is unclear since we are talking about two sets (forget and retain sets). For example, would we call a forget point highly correlated with retain set points if most of its "mass" lies in the space spanned by retain set points or do we care about pair-wise correlations? The paper would become much stronger if it can come up with a precise hypothesis on the exact notion of correlation, and show in some practical settings that as this notion of correlation increases, gradient ascent based methods start to perform poorly.

Nevertheless, the theoretical examples still seem illuminating. Therefore, I am increasing my rating to weak accept.

**Limitations:**

Yes

**Quality:**

3

**Strengths And Weaknesses:**

Gradient ascent based methods are prevalent in machine unlearning. Therefore, any attempt at understanding its weaknesses is welcome.

The main claims/contributions of this paper are:
1. Gradient ascent based methods fail in practical settings and can even degrade model performance.  Moreover, they are very sensitive to choice of hyperparameters. This was shown using experiments on CIFAR.
2. Gradient ascent based methods fail due to correlations between retain and forget set points. This was shown theoretically or small experiments in various stylized settings.

The first claim is largely known from prior work. For example, Georgiev et. al. [1] already show that gradient ascent based methods do not perform well and can also hurt model performance (e.g. Figure 2, section 4 and Figure 11, section 6) . The authors do cite this work.

The second claim, which is the main contribution of the paper, is shown in settings too stylized. For example, one of the settings assumes logistic regression with orthogonal points and further stringent assumptions on how the points are labeled. Firstly, it is unclear how much of the claim relies on the points being correlated. It seems possible that gradient ascent can hurt even in the absence of correlations. As an example of this, see Figure 11 in [1] where gradient ascent starts to hurt model performance eventually (the setting here is under-determined regression with minimal correlations).  Even if correlation causes additional issues, significantly more convincing and thorough demonstration of that is needed, both empirically and theoretically.

As another example, consider the result where it is shown that doing gradient descent on retain set points may not fix the harm done by gradient ascent. This shown for a specific 2 dimensional dataset, fitted using 2 parameters. Again it is unclear how general this observation is. For example, do we expect it to hold for overparameterized neural networks?

Finally, the paper also uses imprecise or unclear language at many places, which makes it hard to understand. For example:
- Line 173: "The data is separable into orthogonal sets Sj for each coordinate j.". While the meaning of this became clearer later, its unclear what orthogonal sets for each coordinate mean.
- Line 193:  "Lemma 3 shows that the DA solution is always farthest away from the
	oracle solution". What does farthest away mean?
- Corollary 1: What is e_j? Also, the text says it follows from lemma 4, but lemma 4 shows no dependence on lambda.
- For lemma 4, is there any condition on alpha? The upper bound seems to become negative when alpha < |F| / |R|?


[1] https://arxiv.org/abs/2410.23232

---

> ### Author Rebuttal · Authors · 2025-07-31
>
> We thank the reviewer for their detailed feedback and the opportunity to clarify our work. We understand the concerns raised regarding novelty, the theoretical setup, and clarity. Below, we address each point in the order in which it was raised. We hope these clarifications will demonstrate the value and significance of our contributions and lead the reviewer to kindly reconsider their assessment and score.
>
> ### Weaknesses
>
> #### **1. On the novelty of Claim 1 (DA methods fail in practice)**
>
> We agree with the reviewer that prior works, such as Georgiev et al. [1], have empirically observed the existence of instabilities in unlearning with DA methods. However, our primary contribution is **not the observation of failure itself**, but the identification and rigorous analysis of a key **underlying cause**: the statistical dependence between forget and retain sets.
>
> Prior work has **not** established this causal link. Our contributions here are twofold:
>
> *  **Empirical Disentanglement:** We deliberately designed experiments to isolate the effect of correlation. In **Fig. 2**, we show that for random sets (where correlations are present but not systematically structured), DA methods are highly unstable, sporadically succeeding but often breaking the model. In contrast, in **Fig. 3**, where we unlearn sets with high structural correlation (points aligned with top PCs of the influence matrix), the failure is systematic and complete. To our knowledge, this direct empirical demonstration linking the *degree of correlation* to the *severity of failure* has not been demonstrated in the existing literature. On top of this, our work shows fine-grained results considering all parameter settings while [1], and other previous works, evaluate over a grid of hyperparameters and report the best score relative to compute time (Appendix C3 in [1]) which can mislead practitioners as explained in the "Ascent Forgets Illusion" (Figure 2).
>
> *   **Distinct Findings from Georgiev et al. [1]:** The specific figures cited by the reviewer show different phenomena than what we analyze.
>     * Figure 2 in [1] shows that different points unlearn at different rates for a fixed hyperparameter setting of a DA method (SCRUB).
>     * Figure 11 in [1] shows that for a *linear* model, gradient ascent **eventually** diverges, which one might argue can be fixed with early stopping.
>     * Our work goes further by providing a theoretical explanation as to *why* this happens in more complex, non-linear problems (logistic regression) and demonstrating a more severe failure mode: DA can be **actively detrimental from the very first step**, regardless of early stopping (Lemma 3).
>
> #### **2. On the theoretical settings being "too stylized" (Claim 2)**
>
> We appreciate the reviewer's concern about the generality of our theoretical results. Our goal with these "stylized" settings was to use a standard fundamental scientific approach: isolate a complex phenomenon (the effect of correlation in DA unlearning) in a tractable environment where it can be analyzed with mathematical rigor.
>
> *   **Clarification on the "Orthogonal" Setting:** We thank the reviewer for highlighting this, as it seems to be a key point of confusion. The data in our analysis is explicitly **non-orthogonal** along the dimensions of interest, which is central to our claim. The "semi-orthogonal" construction (Assumption 1) means that we are analyzing a high-dimensional space where dependencies are localized to specific coordinates. This allows us to prove that correlations along a few dimensions can cause failure, *even when all other data dimensions are perfectly orthogonal and non-interfering*. This setup strengthens, rather than weakens, our claim by showing that cross-dimensional information cannot necessarily rescue the unlearning process and save the model. We would stress that in purely orthogonal data, DA unlearning may not be problematic, but this is a case of less relevance to the real world.
>
> *   **On the "Stringent" Labeling Assumption:**
>  The only additional labeling assumption made throughout the paper is Assumption 2. It is easy to see that the assumption is rather general and it simply requires that no two points, one in the forget set other in the retain set, with the same data (x) have opposite labels (y). This is a requirement to establish that **no step size or stopping criteria** can salvage the unlearning process when DA is used in this scenario. It is a mild assumption in practice since on most datasets (e.g. cifar10, imagenet) very similar data points rarely have opposite labels, as one might expect.
>
> *   **On Correlation as the Cause of Failure:** The reviewer suggests DA might hurt even without correlations, citing Fig. 11 in [1]. While other failure modes may exist, our work's focus is to prove that **correlation is a sufficient condition for catastrophic failure**. Our Lemma 3 demonstrates that for our non-linear setting, the DA solution *progressively moves away* from the oracle solution from the very beginning, for *any* step size, making the process actively harmful regardless of early stopping. This is a much stronger and more specific result than observing eventual divergence in a linear model, which one could argue can potentially be resolved with early stopping.
>
> #### **3. On the generality of the 2D "Bad Minima" example**
>
> The 2D example in Section 5.2 is intended as a clear, visualizable illustration of a more general mechanism: **how DA can maliciously reshape the loss landscape to trap a model in a suboptimal basin of attraction.**
>
> The reviewer asks if this holds for overparameterized neural networks. While a full analysis is beyond the scope of our paper, the underlying principle remains valid. In any model, non-convex or otherwise, the final parameters are a result of balancing gradients from all data points. If a forget set is highly correlated with a subset of the retain set that is critical for defining a good decision boundary (e.g., analogous to support vectors), the ascent-descent process does more than just "remove" information. It can create a "hole" in the loss landscape where the correct minimum used to be, while simultaneously deepening an incorrect minimum elsewhere. This trapping mechanism is not exclusive to low dimensions and provides a strong intuition for why simple finetuning on the retain set afterwards may fail to recover the correct solution. As a concrete example, consider our setting of semi-orthogonal data with a smaller number of data points than dimensions $d>N$, where there are still multiple points correlated on a single axis. Since the data is orthogonal on the other axes, unlearning is still effectively a one dimensional problem, although the network operates in the overparameterized regime, and our results hold. We agree that repeating the analysis for more complex overparameterized networks is an excellent natural extension, which we postpone to future work. As a final point we would like to argue that the regime with $N > d$ has become more common in practice, where our findings clearly apply, for example, Large Language Models are usually trained on many more tokens than their number of parameters.
>
> ### Questions & Clarifications
>
> We thank the reviewer for pointing out several areas of unclear language. We will revise the paper to improve clarity based on this feedback.
>
> *   **Line 173: "The data is separable into orthogonal sets Sj for each coordinate j."**
>     *   **Clarification:** We will rephrase this to be clearer. This assumption simply means the data matrix has a block-diagonal structure after a permutation of rows, where each block corresponds to a set of samples that only have non-zero elements on a unique, shared set of coordinates. Thus, samples in different sets $S_j$ and $S_k$ (for $j \neq k$) are orthogonal, while samples within the same set $S_j$ can be (and are in our analysis) correlated.
>
> *   **Line 193: "Lemma 3 shows that the DA solution is always farthest away from the oracle solution".**
>     *   **Clarification:** In the 1D context of Lemma 3, "farthest away" has a precise geometric meaning that we will make explicit. It means the DA solution $w^{\text{DA}}$ and the oracle solution $w^{\mathcal{R}}$ lie on *opposite sides* of the original model's solution $w^{\mathcal{D}}$. Therefore, any step in the DA direction moves the model *away* from the oracle, making it a worse solution than keeping the pretrained model as is.
>
> *   **Corollary 1: "What is e_j? Also, the text says it follows from lemma 4, but lemma 4 shows no dependence on lambda."**
>     *   **Clarification:** We apologize for the typo; $e_j$ should be $\alpha$. We thank the reviewer for pointing out this error. The corollary follows from combining **Lemma 4 and Lemma 5**. Lemma 4 shows the distance between the original and oracle solutions is bounded (independent of $\lambda$). Lemma 5, its counterpart, provides a lower bound on the distance between the DA and oracle solutions, which *grows* as $\lambda \to 0$. The combination of these two results leads directly to the conclusion of the corollary, which is a divergence of the solutions. We will correct the text and the typo.
>
> *   **Lemma 4: "Is there any condition on alpha? The upper bound seems to become negative when alpha < |F| / |R|?"**
>     *   **Clarification:** The right-hand side of the inequality is non-negative because the logarithm is **inside an absolute value**: $|\ln((1+\alpha))|$. This may not have been immediately obvious. We thank the reviewer for this close reading and will add a note to the text to prevent this confusion. The bound holds for any $\alpha > 0$.
>
> [1] Georgiev et al "Attribute-to-Delete: Machine Unlearning via Datamodel Matching." arXiv:2410.23232v2

---

> > ### Comment · Reviewer_85tp · 2025-08-07
> >
> > I thank the authors for the detailed response. Most of my questions on the relevance of the analysis are answered. It would be great to add some of the discussion in the response to the next version of the paper, and also improve the clarity of the writing.
> >
> > The part that still seems a bit lacking is the empirical demonstartion in that there is not enough convincing demonstration that correlations are a major reason behind the faliure of gradient ascent based methods in practice. Also, the precise definition of correlation when we move away from stylized settings is unclear since we are talking about two sets (forget and retain sets). For example, would we call a forget point highly correlated with retain set points if most of its "mass" lies in the space spanned by retain set points or do we care about pair-wise correlations? The paper would become much stronger if it can come up with a precise hypothesis on the exact notion of correlation, and show in some practical settings that as this notion of correlation increases, gradient ascent based methods start to perform poorly.
> >
> > Nevertheless, the theoretical examples still seem illuminating. Therefore, I am increasing my rating to weak accept.

---

> > > ### Author Response · Authors · 2025-08-08
> > >
> > > We are grateful to the reviewer for their thoughtful engagement with our response and for raising their score. We sincerely appreciate their feedback and will ensure the clarifications discussed are integrated into the final version of the paper.
> > >
> > > The reviewer raises an important point regarding the empirical demonstration and the precise definition of "correlation" in complex, real-world settings. We agree that this is highly non-trivial. Our response is two-fold:
> > >
> > > 1) We acknowledge that rigorously defining and quantifying the **most** detrimental forms of correlation in practice, be it pairwise similarity, projection onto the subspace spanned by the retain set, or another metric, is a challenging and important open research question. Our work's primary objective was to first provide a definitive theoretical proof that such dependencies represent a fundamental failure mode for DA methods, a fact that was previously only hinted at empirically. We aimed to establish the idea of: "if correlation exists, then failure can occur", which we hope will motivate future work.
> > > 2) To build on this basic concept, we believe the reviewer's suggestion is a natural next step. As a potential direction, we propose that the cross-influence matrix between the forget and retain sets could serve as a powerful and tractable proxy for quantifying these dependencies. The influence function framework is well-suited for measuring how a model's predictions on the retain set are affected by the presence of the forget set, directly capturing the interplay we are concerned with. One could then empirically test the hypothesis that as a tractable metric derived from this matrix increases (e.g., its spectral norm), the performance of DA unlearning systematically degrades. We believe this would make for a very interesting follow-up study, which we have also considered.
> > >
> > > We thank the reviewer once again for their constructive criticism, which has helped improve our current paper and align our future directions with the interests of others in the community.

---

### Official Review · Reviewer_s9Rb · 2025-07-02

**Clarity:** 4
**Significance:** 3
**Originality:** 3
**Rating:** 5
**Confidence:** 3

**Summary:**

The paper studies gradient-ascent/decent (DA) machine unlearning methods that rely on taking gradient ascent steps on a forget set while taking gradient descent steps on a retain set.
It demonstrates empirically that DA can fail when the forget set points are chosen to be highly correlated with the original set.
For randomly chosen forget sets, it finds that such methods fail to forget while degrading model performance, arguing that circumstances within which they do seem to work require access to privileged information for choosing hyperparameters that is typically unavailable.
It further theoretically studies to what extent statistical correlations between data points cause DA methods to fail.
To this end, it considers a logistic regression model on datasets with specific correlation structure between data points.
It finds that when samples are fully correlated, the DA solution and the oracle solution lie in opposite directions with respect to the pretraining solution.
As a consequence any amount of DA unlearning in this setup degrades both model performance and unlearning performance.

**Questions:**

- To what extent are strong statistical correlations between the forget and retain dataset an ecologically valid assumption in the context of unlearning? I would expect that the typical example of unlearning specific facts, such as a person's telephone number, does not fit this assumption very well?

- Continuing this line of thought, the choice of unlearning CIFAR-10 data points strikes me as odd as the correlations of individual data points with the whole dataset might be relatively high simply due to the small size of the dataset. Would you expect the problems of DA due to correlations to be less prevalent as the scale of the data increases and the fraction F/D decreases as a result?

## Minor

- Typo in L142: [...] would be indistinguishable (to?) those of the test [...]
- Typo in L165: [...] in low-dimensions ~with~ on a small fixed dataset.

**Ethical Concerns:**

["NO or VERY MINOR ethics concerns only"]

**Final Justification:**

- The paper is well presented and structured
- The research question of how well DA-based methods work in the presence of correlations between the forget and retain set, is well motivated
- Both the empirical and theoretical setup are well designed
- The findings have significant implications for the applicability of DA machine unlearning methods

**Limitations:**

Limitations have been adequately addressed.

**Paper Formatting Concerns:**

None.

**Quality:**

3

**Strengths And Weaknesses:**

The paper is clearly written with an easy to follow structure. The research question of how well DA-based methods work in the presence of correlations between the forget and retain set, is well motivated. Both the empirical and theoretical setup are well designed and the findings have significant implications for the applicability of DA machine unlearning methods.

_Please note that I cannot particularly comment on the novelty of the findings since I am not an expert in the field of machine unlearning and do not know the related work very well._

---

> ### Author Rebuttal · Authors · 2025-07-31
>
> We would like to thank the reviewer for their work and the positive evaluation of our work.
>
> Below, we address the specific questions raised in the review.
>
> ### Questions:
>
> - **Are strong statistical correlations between the forget and retain dataset an ecologically valid assumption (unlearning telephone numbers etc.) :**
> While unlearning a specific fact like a phone number might seem like a low-correlation task, subtle dependencies still exist. A phone number is not a random string; its components, such as the area and central office codes, are strongly correlated with geographic location. If the model is given the associations between a person and their city/state (which would be in the retain set), the area code in the forget set is not independent. Our work shows that even such nuanced correlations, which are often overlooked, can be sufficient to cause DA methods to fail.
> More importantly, many of the most prevalent and large-scale unlearning requests are inherently high-correlation problems. Consider:
>     - Unlearning Codebases: A frequent request involves removing a proprietary codebase from a Large Language Model. The "forget set" (the proprietary code) will inevitably share common libraries (e.g., NumPy, TensorFlow), widely-used algorithms, and standard design patterns with the "retain set" (e.g., public code from GitHub). The statistical dependence here is unavoidable.
>     - Unlearning Visual Data: Imagine an autonomous vehicle model trained on footage from both public and private roads. If a request is made to unlearn footage from a specific private property, this "forget set" is visually almost indistinguishable from the "retain set" of public roads. Both contain asphalt, trees, lane markings, and curbs. Attempting to unlearn one via gradient ascent risks degrading the model's fundamental concept of what a "road" is, a prime example of the detrimental failure modes we analyze.
>     - Unlearning Copyrighted or Stylistic Content: In generative models, a request might be to unlearn a specific artist's work. The "forget set" (the artist's images) is highly correlated with the "retain set" through a shared artistic style, subject matter, or visual motifs that the model has learned to generalize. Forgetting the specific instances without harming the stylistic understanding is precisely the challenge our work addresses.
>
>
> - **Cifar 10 datset:**
> Cifar 10 is rather standard for testing in Machine Unlearning (e.g [1-4]). The dataset is not degenerately correlated as MNIST or Fashion-MNIST (please see the response we give to question 3 asked by reviewer MLpH). Pretrained models are clearly distinguishable from oracle models trained from scratch without the forget set by a notable divergence in the forget set margins distribution (high $95^{\text{th}}$ percentile of KLoM scores, e.g. grey dotted vertical line in Figure 1).
>
> - **Scale of dataset and F/D:**
> Our results in section 5 crucially rely on the fraction of $|\mathcal{F}|/ |\mathcal{D}|$. The overall size of the dataset is not important in our derivations. Nevertheless, it is true that as the fraction of forget points out of the entire dataset $|\mathcal{F}|/ |\mathcal{D}|$ goes to 0, where the model is asked to unlearn a negligible fraction of its training data, DA may not necessarily be harmful, as shown in section 5.1. In the limit of $|\mathcal{F}|/ |\mathcal{D}| \to 0$, it is also important to note that the differences between the pretrained model and the oracle would collapse so it is likely that a DA based method that just stays very close to the pretrained model would achieve reasonable performance.
>
> - **Minor Typos:**
> We thank the reviewer for pointing out the typos, we have corrected them in the revised manuscript.
>
> [1] Georgiev et al "Attribute-to-Delete: Machine Unlearning via Datamodel Matching." arXiv:2410.23232v2
>
> [2] Kurmanji et al. "Towards Unbounded Machine Unlearning." arXiv:2302.09880
>
> [3] Hoang et al. "Learn to Unlearn for Deep Neural Networks: Minimizing Unlearning Interference with Gradient Projection." arXiv:2312.04095v1
>
> [4] Goel et al. "Towards Adversarial Evaluations for Inexact Machine Unlearning." arXiv:2201.06640v3

---

> > ### Comment · Reviewer_s9Rb · 2025-08-02
> >
> > Thank you for your insightful answer. My questions have been addressed and I maintain my recommendation for acceptance of this paper.

---

### Official Review · Reviewer_oUA1 · 2025-07-03

**Clarity:** 2
**Significance:** 2
**Originality:** 3
**Rating:** 4
**Confidence:** 2

**Summary:**

This paper studies the problem of machine unlearning. It first empirically shows that descent-ascent-based methods tend to fail at machine unlearning tasks under robust evaluation. They present a theoretical argument showing that the statistical dependencies between the forget set and the retain set are the reason for the observed phenomenon.

The paper provides a theoretical analysis of multi-dimensional logistic regression, showing inter-set correlations lead to DA failure modes.

The paper also provides a qualitative analysis of a binary 2-dim classification example with a simple network (a sigmoidal network with two parameters), showing how DA leads to sub-optimal solutions that is different from the retrained model.

**Questions:**

Q1. Regarding Lemma1, is Hoeffding’s Inequality applied to the model predictions? When forgetting set is used for the training, the predictions of the trained model are dependent on the training sample (forgetting set). This seems to contradict the assumptions of Hoeffding’s Inequality.

Q2. Does the result in Section 5.2 require any assumption on the learning rate? e.g. equal learning rate in ascent and descent?

**Ethical Concerns:**

["NO or VERY MINOR ethics concerns only"]

**Final Justification:**

The rebuttal clarifies my concern regarding lemma 1. Therefore, I raise my score from 3 to 4. However, it should be noted that since my expertise does not lie in machine unlearning, I cannot fully evaluate the novelty and the impact of this work.

**Limitations:**

yes

**Paper Formatting Concerns:**

The main text of the paper is also included in the supplementary material.

**Quality:**

2

**Strengths And Weaknesses:**

Strengths:
- The paper conducts a theoretical analysis in the multi-dimensional logistic regression problem to investigate how the relationship between forget and retain sets affects machine unlearning performance.
- The qualitative analysis using a simple network is illustrated, showing how the unlearned model differs from the retrained model
- Figure 2 shows an illusion that DA-based method is mistakenly considered successful. This may be useful for the community.

Weakness:
- Lemma 1 seems problematic and does not consider the effect of model training. Hoeffding’s Inequality requires the random variable to be independent. Based on Appendix B, Hoeffding’s Inequality is applied by taking model prediction as a random variable, i.e. $ 1_{(h_\theta(x_i), y_i) }\in[0,1] $ (line 796). As the forget set is used in the gradient ascent, the final predictions with trained model parameters are dependent on all the training samples.

- In Figure 5 and section 5.2, there is no discussion on the effect of training dynamics, e.g. learning rate or regularization parameter $\lambda$. Does the conclusion depend on any assumption about the learning rate?

- It would be better to include more discussion on the related works or future directions regarding how to deal with of data dependencies between the forget and the retain sets in algorithm design.

---

> ### Author Rebuttal · Authors · 2025-07-31
>
> We thank you for your detailed and constructive review. We appreciate that you found our contribution to be useful to the community.
>
> Below, we address the specific weaknesses and questions brought up in your review.
>
> ### Weaknesses:
>
> - **Lemma 1 seems problematic and does not consider the effect of model training.**
> We apologize for this crucial misunderstanding and would like to clarify the confusion regarding Lemma 1. In the text above the Lemma, we briefly state that this Lemma corresponds to the **oracle** model, trained only on the retain set, and **not to a pre-trained and then unlearned model**. It is important to stress that the **oracle** model is obtained by training from scratch on the retain set, and so it is **never** exposed to the forget set. Therefore, there are no dependencies between the random variables since the model parameters are only dependent on the retain set. We will clarify this further by specifying in the statement of the Lemma that we refer to the **oracle model** instead of simply the **model** as in the current version.
>
> - **In Figure 5 and section 5.2, there is no discussion on the effect of training dynamics.**
> Our results throughout section 5 concern the landscape of the optimization process and the locations of the optimal minimizers. As a result, the training dynamics do not affect them, as long as the pre-defined objective function is unchanged. For example, the magnitude of the learning rate is not important and no assumptions have been made on it to draw the conclusions for the solutions of the optimization problem. Regarding the regularization parameter, similar to section 5.1.1, changing it does not alter the relative positions of the minimizers, it only controls their distance from the origin and between them. Hence, the decision boundaries in Figure 5 are not affected by the regularization parameter.
>
> - **It would be better to include more discussion on the related works or future directions.**
> Thanks for the suggestion, we would be happy to include more details in the future directions and related work for the camera ready version.
>
> ### Questions:
>
> - **Question 1: Hoeffding's inequality:**
> Regarding the application of Hoeffding's inequality, it is indeed applied to model predictions. Nevertheless, as we mentioned in our reply to the raised weaknesses, Lemma 1 concerns the **oracle model** which has been trained using only the **retain** set and has **not** seen the **forget** set, so it does not contradict the assumptions required for the inequality to hold.
>
> - **Question 2: The result of section 5.2 and equal learning rates:**
> We thank the reviewer for this very interesting question. As we mentioned above, the learning rate magnitude has no real effect on our results and they would still hold. However, selecting different learning rates for the retain and forget sets (i.e. the ascent and descent steps on DA) alters the objective function. Specifically, the new objective function would be weighted according to the ratio of the learning rates. While it is possible to select a weighting so that the specific example in Figure 5 is not problematic, one could find a problematic instance along the lines we describe for any ratio. For example, assume that for the unlearning process we select a step size for the forget set that is double that of the retain. Now, in the Figure 5 instance you can simply reduce the size of the forget set from 2 samples to 1 (by removing the other sample from the original dataset), this would lead again to the same optimization landscape as the one we have described.
>
> We sincerely hope that you kindly consider raising your score if our reply has alleviated your concerns regarding our work.

---

> > ### Comment · Reviewer_oUA1 · 2025-08-04
> >
> > Thank you for your time in writing the detailed rebuttal. I appreciate the clarification regarding Lemma 1, and I found the discussion around the equal learning rate particularly interesting. I still have a quick question about Lemma 1.
> >
> > Since Lemma 1 is stated for the oracle model, it seems to me that it may not have a direct connection to the gradient-based unlearning method. Could the authors provide more context on the purpose of including Lemma 1 in the paper?
> >
> > In particular, I’m a bit confused by the statement below Lemma 1 (line 150):
> >
> > >Lemma 1 suggests that methods based on DA that degrade a metric on the forget set, might be more harmful rather than beneficial.

---

> > > ### Author Response · Authors · 2025-08-04
> > >
> > > Thank you for your response. Below we provide additional clarification regarding the phrasing and intention of Lemma 1.
> > >
> > > Unlearning methods, both ascent-based and ascent-free, aim to produce a model that is statistically similar to an oracle. For a successful unlearning method, **the unlearned model and oracle model should have similar accuracies on all sets**. Note that similar accuracies is a consequence and a weaker condition than having high statistical similarity or low KLoM in practise. Lemma 1 shows that for a random forget set **oracle accuracies on the forget set and test set are close**.
> > >
> > > Under proper training, the oracle should have relatively **high accuracy on the test set and, due to Lemma 1, on the forget set**. Then, any unlearning method that **focuses on degrading accuracy on the forget set** will have a **different accuracy than the oracle** model on this set, **breaking the unlearning goal** of achieving a model statistically similar to a fully retrained one (oracle).
> > >
> > > To summarize the logic is: any unlearning method should approximate the perfect oracle $\text{Unlearning}\to \text{Oracle}$. The perfect oracle should treat a random forget set on equal footing with the test set $\text{Oracle}(\text{Forget}) \sim \text{Oracle}(\text{Test})$ (Lemma 1). Therefore, if $\text{Unlearning}(\text{Forget}) \neq \text{Unlearning}(\text{Test})$ then the unlearning method has failed.
> > >
> > > We will rephrase line 150 as follows:
> > > "Lemma 1 suggests that any unlearning method which approximates the perfect oracle should result in a model which performs equally well on both the forget and the test set. Therefore, an unlearned model with poor forget set performance will statistically diverge from an oracle model, whose forget set accuracy reflects the accuracy on the test set and is high for modern machine learning techniques."
> > >
> > >
> > > We would be happy to discuss further if there are any remaining questions left unresolved.

---

> > > > ### Comment · Reviewer_oUA1 · 2025-08-06
> > > >
> > > > Thank you for taking the time to provide additional clarifications. I revisited the paper and other reviews. I find the paper’s findings quite interesting and have raised my score to 4. However, it should be noted that since my expertise does not lie in machine unlearning, I cannot fully evaluate the novelty and the impact of this work.

---

> > > > > ### Author Response · Authors · 2025-08-08
> > > > >
> > > > > We sincerely thank the reviewer for their time and for raising their score. We are pleased that our clarifications were helpful and that they found our paper's findings interesting.

---

### Decision · Program_Chairs · 2025-09-17

**Decision:**

Accept (poster)

**Comment:**

This paper studies the (failure of) machine unlearning problem in the presence of high correlation between the forget and retain data sets, and itproposes both experimental analyses and theoretical justifications to such behavior. Reviewers have in general find this work interesting and timely, and have reached a consensus on acceptance -- the AC agrees and recommend accept.